# *Drosophila* Glia: Models for Human Neurodevelopmental and Neurodegenerative Disorders

**DOI:** 10.3390/ijms21144859

**Published:** 2020-07-09

**Authors:** Taejoon Kim, Bokyeong Song, Im-Soon Lee

**Affiliations:** Department of Biological Sciences, Center for CHANS, Konkuk University, Seoul 05029, Korea; jusink@konkuk.ac.kr (T.K.); bokysong@konkuk.ac.kr (B.S.)

**Keywords:** glia, glial defects, *Drosophila* models, CNS disorders

## Abstract

Glial cells are key players in the proper formation and maintenance of the nervous system, thus contributing to neuronal health and disease in humans. However, little is known about the molecular pathways that govern glia–neuron communications in the diseased brain. *Drosophila* provides a useful in vivo model to explore the conserved molecular details of glial cell biology and their contributions to brain function and disease susceptibility. Herein, we review recent studies that explore glial functions in normal neuronal development, along with *Drosophila* models that seek to identify the pathological implications of glial defects in the context of various central nervous system disorders.

## 1. Introduction

Glial cells perform many important functions that are essential for the proper development and maintenance of the nervous system [1]. During development, glia maintain neuronal cell numbers and engulf unnecessary cells and projections, correctly shaping neural circuits. In comparison, glial cells in the adult brain provide metabolic sustenance and critical immune support. Thus, the dysfunction of glial activity contributes to various central nervous system (CNS) disorders in humans at different stages of life [2]. Accordingly, the need for research regarding the initiation as well as the progression of disorders associated with glial cell dysfunction is increasing.

The proportion of glia in the nervous system has reportedly increased in favor of glia during the course of evolution [3,4,5,6]. However, *Drosophila* glial cells still display a very similar morphology and molecular functionality to their mammalian counterparts. This suggests that investigations into fly glia have the potential to enrich our understanding of the fundamental aspects of glial cell biology. As the nervous system of *Drosophila* is simpler than that of humans, modeling human brain diseases in *Drosophila* offers numerous advantages for elucidating the molecular and cellular mechanisms underlying the complicated diseases caused by glial dysfunction. In addition, approximately 77% of human disease-associated sequences in the Online Mendelian Inheritance in Man (OMIM) database strongly match sequences in the *Drosophila* sequence database [6]. Furthermore, as *Drosophila* provides a number of excellent molecular-genetic tools to study a diverse array of glia–neuron interactions, it is considered one of the best model organisms for providing exciting insights into various glial dysfunctions in human CNS disorders.

CNS disorders are categorized into two groups depending on the disease onset time during brain development: neurodevelopmental and neurodegenerative. Neurodevelopmental disorders arise due to damage or abnormal development of the brain at an early age. These disorders include autism, learning disabilities, and various developmental delay disorders [7]. In contrast, neurodegenerative diseases commonly involve age-related dysfunction of neuronal maintenance over a lifetime and are caused by the progressive loss of specific neuronal populations [8]. Neurodegenerative diseases are commonly associated with the formation of cellular aggregates of toxic proteins, representative examples of which include Huntington’s disease (HD) and Alzheimer’s disease [9]. Although for many years these CNS disorders were thought to result exclusively from the aberrant function of neurons, numerous pieces of recent evidence suggest that glial cell dysfunction is involved in a variety of brain diseases, as it affects the dynamics of neuron and glia networks [10]. Thus, in this review, we summarize recent progress in the research of glial contribution to brain function and CNS disease susceptibility using *Drosophila* disease models.

The main part of this review is divided into three sections, describing subtypes of glia in the *Drosophila* CNS during development, glial cell function in *Drosophila*, and *Drosophila* models of human neurodevelopmental and neurodegenerative CNS disorders associated with glial dysfunction.

## 2. Subtypes of Glia in *Drosophila* CNS during Development

Glial development in *Drosophila* has been well studied from embryo to adult. *Drosophila* CNS glial cell progenitors are first formed during the embryonic stages. The majority of the embryonic CNS glial cells, the lateral glia, derive from the neurogenic region of the ectoderm [11], while a unique subset of embryonic glia, midline glia, are of mesectoderm origin [12]. The lateral glial cells are produced from neuroglioblasts or glioblasts that generate mixed lineages of neurons and glial cells or glial progeny only, respectively [11]. In the presence of the glial cell missing (Gcm) transcription factor and its related factor, Gcm2, which act as fate determinants, the lateral glial cells are specified according to positional information from the neuroectoderm [13]. In the lateral glial cells, Gcm activates a set of downstream transcription factors that control the differentiation and maintenance of the glial cell fate. Among the target genes of Gcm, the reversed polarity (Repo) homeodomain transcription factor promotes the proteasome-mediated degradation of Gcm and positively regulates its own promoter, resulting in its sustained expression in glia. Together with Repo, other Gcm-induced transcriptional factors also play crucial roles in glial specification, such as locomotion defects (Loco), pointed (Pnt), and tramtrack (Ttk). Pnt and Loco function as activators of glial fate [14,15,16,17], while Ttk is a repressor of neuronal fate [18]. Thus, neuroblasts from the neuroectoderm appear to have an inherent primary fate to develop as neurons, and a neuronal fate is chosen in the absence of Gcm. The lateral glial cells in the *Drosophila* embryonic CNS have been assigned to three subtypes according to their spatial distribution and morphology: the surface-, the cortex-, and the neuropil-associated glial cells [19]. Although *Drosophila* gliogenesis begins in the embryo [20,21], important morphogenic changes in glial cells take place during the larval and pupal stages. This overlaps with the second wave of neurogenesis responsible for the formation of 90% of the neurons in the adult CNS [22]. For example, the midline glia disappear in the post-embryonic stages [12], while cortex-associated glial cells infiltrate the entire cortex region of the CNS during early larval stages to establish nonoverlapping spatial domains [23]. More drastic events take place during pupal stages. In the case of neuropil-associated glia, the glial cells born during the embryonic stages persist until the end of the larval stages, eventually undergoing apoptosis during metamorphosis. Later, adult neuropil-associated glial cells are derived from secondary glia precursor cells that derive from Type II neuroblast lineages during early metamorphosis [24]. Herein, we will limit this review to describe the three main glial subtypes that are found in the CNS of larval and adult *Drosophila*: surface-associated, cortex-associated, and neuropil-associated glia.

The *Drosophila* larval CNS consists of the brain hemispheres and the ventral nerve cord (Figure 1). All neuronal cell bodies of the CNS reside in the neuronal cell cortex region, whereas axons and dendrites project to the neuropil (Figure 1A). Surface-associated glia, including perineurial and subperineurial glia (SPG), act as a chemical and physical barrier for the CNS [25,26]. These two glial cell types are found in both the CNS and the peripheral nervous system (PNS) (Figure 1A,B). Perineurial glia form the outermost cellular layer surrounding the CNS and the PNS. From the later larval stages onward, perineurial cells form a discontinuous layer with small gaps [27,28]. In contrast, underneath the perineurial layer, a flattened monolayer of SPG covers the entire CNS surface. During larval development, polyploidization causes an increase in SPG cell size in order to cover the brain [29]. These cells function as a blood–brain barrier (BBB) by forming septate junctions with one another, which is important for keeping high potassium concentrations in the hemolymph separate from the nervous system and for proper action of potential propagation [30,31,32].

The second subtype, cortex-associated glia, is present in the neuronal cell cortex and associates with all neuronal cell bodies, isolating them from other neurons by individually wrapping each neuronal soma. By the late embryonic stage, a single cortex glial cell can surround approximately 100 neuronal cell bodies individually in the CNS and form the honeycomb-like glial membrane structure called the trophospongium [27]. Cortex glial cells associate directly with the subperineurial layer, as well as neuropil-associated glia [19,33,34]. They are also thought to be involved in trophic support for neurons and gas exchange [35].

The third subtype, neuropil-associated glia, includes ensheathing glia and astrocyte-like glia [19]. Ensheathing glia extend flattened processes along the edges of the neuropil to cover the synaptic neuropil. They also subdivide brain lobes and major commissures into histologically discrete compartments [27,36,37]. The cell bodies of astrocyte-like glia also lie in between the cortex and neuropil; however, in contrast to ensheathing glia, their cell processes infiltrate into the neuropil and form a dense meshwork of very fine processes, very close to synapses [38]. Astrocyte-like glia are by far the most extensively studied among all the glial subtypes discussed above. These cells appear to interact with each other and extend processes to cover most of the neuropil synaptic space in the larval and adult nervous systems [38,39].

## 3. Functions of Glial Cells in *Drosophila*

### 3.1. General Comparisons between Drosophila Glia and Mammalian Glia

The glial cells in the *Drosophila* nervous system are considerably simpler but distinct from their mammalian counterparts. For example, *Drosophila* glia comprise only approximately 10%–15% of the 90,000 cells estimated to arise in the adult CNS, whereas mammals contain more glia: 50% in mice and 90% in humans. This suggests an expansion of glial roles, such as learning, as the complexity of the nervous system increases [3,4,5,40]. In addition, while axons in bundles are associated with glia in *Drosophila*, axons in vertebrates are individually ensheathed by Schwann cells or oligodendrocyte-like myelin sheaths [31,41]. Furthermore, microglia, the only immune cells that reside in the brain, are present in mammals but not in *Drosophila*. Microglia play essential roles in pruning and maintaining the synapses; however, in *Drosophila*, the ensheathing glia are instead responsible for engulfing degenerating axons [37]. Mammalian microglia are derived from embryonic mesoderm in the yolk sac [42,43,44]; this contrasts with macroglia (astrocytes and the oligodendrocyte lineage) that arise from neuroepithelial progenitor cells in the embryonic neural tube and forebrain [45].

Nevertheless, the *Drosophila* nervous system also has numerous features in common with its mammalian counterpart [46]. The molecules mediating electrical excitability and related channels in *Drosophila* are generally similar to those in humans, including voltage-activated K^+^ channels [47]. In the case of *Drosophila* glia, a number of evolutionarily conserved features have been identified in its morphology as well as in its function, such as the electrical insulation and metabolic support of neurons, the modulation of neuronal activity by regulating neurotransmitter homeostasis, or the direct influence on neuronal activity via gliotransmitters [2]. *Drosophila* astrocyte-like glial cells tile with one another to establish unique spatial domains, just like their vertebrate counterparts [38,48].

In the following section, we describe the functions of *Drosophila* glial cells in detail.

### 3.2. Functions of Glial Cells in the Drosophila Nervous System

#### 3.2.1. Development and Maintenance of Neural Circuits

Glial cells have important functions in neural development. As was initially demonstrated in *Drosophila* [49], including in its embryonic CNS [50,51,52], some glial cells perform key functions in axon guidance [49]. Later, it was shown that signals between glia and neurons are reciprocally required for the *Drosophila* visual system to develop properly; axons need glia to depart from the eyestalk, while glia require signals from neurons to migrate back along axons [53].

During development, glia maintain neuronal cell numbers and ensure that mature neural circuits are correctly shaped by engulfing unnecessary cells and neurites. Glial cells are the main phagocytes in the CNS, performing their phagocytic function by engulfing dying neurons and surplus neuronal branches and synapses and by degenerating neuronal axons [43,54,55,56,57,58]. Throughout the embryonic neurogenesis of *Drosophila*, approximately 30% of all cells undergo programmed cell death and are engulfed by glia [59,60,61,62]. Two transmembrane glial phagocytic receptors, SIMU (homolog of the mammalian Stabilin-2) and Draper (homolog of the mammalian MEGF10 and Jedi), are known to mediate the glial phagocytosis of apoptotic neurons. However, during metamorphosis, Draper, but not SIMU, plays a critical role in the glial clearance of apoptotic cells via c-Jun N-terminal kinase (JNK) signaling, predominantly by ensheathing glia and astrocyte-like glia [63]. In addition, the glial phagocytic ability helps to mediate the dramatic reorganization of neural circuits in the larval CNS into the more complex adult brain at metamorphosis [64]. Furthermore, astrocyte-like glial cells mediate the removal of neuronal debris that accumulates during axonal pruning in the mushroom bodies (MB) by clearing axonal debris via Draper, as well as by promoting MB axon fragmentation [65,66,67,68]. Glial phagocytic ability declines with age; this correlates with neuronal dysfunction, suggesting that increased glial phagocytosis may prevent neurodegeneration. However, in opposition to this hypothesis, Hakim-Mishnaevski et al. (2019) recently reported that neurodegeneration occurs with both defective and excessive glial cell function. This highlights the fact that the tight regulation of glial phagocytosis is important for proper brain function [69].

#### 3.2.2. The Blood–Brain Barrier

The BBB separates the nervous system from the blood. The mammalian BBB is established by the interactions between endothelial cells, pericytes, and astrocytes [70]. Signals from pericytes induce endothelial cells to form tight junctions that prevent the paracellular diffusion of ions and nutrients from the bloodstream into the brain parenchyma. Astrocytes play only a modulatory role in this barrier [71,72,73,74].

Unlike the BBB in mammals, the *Drosophila* nervous system is separated from the hemolymph, the circulating fluid that is analogous to vertebrate blood, by SPG that block paracellular diffusion [32,75,76,77]. The specialized septate SPG junction is similar to the specialized tight junction complexes of endothelial cells seen in the vertebrate BBB [77,78]. Interaction with other glial cell types, such as perineurial glial cells, has been demonstrated to regulate the size selectivity of the *Drosophila* BBB [28].

#### 3.2.3. Electrical Insulation of Axons

In vertebrates, oligodendrocytes in the CNS and Schwann cells in the PNS produce multilayered myelin membranes around axons, providing distinct domains that are essential for the conduction of saltatory action potentials [79,80,81]. In *Drosophila*, ensheathing glia encase the neuropil or axon fascicles of all CNS cells, whereas wrapping glial cells wrap individual axons in the PNS (Figure 1A, B). The structure of these wrapping glial cells is similar to Remak bundles, unmyelinated axon bundles in the mammalian PNS [28,82,83,84]. Although there is no myelin in *Drosophila*, glial “hyper wrapping” around individual axons can be observed in the CNS of adult flies lacking the swiss cheese (*sws*) gene [85,86]. It can also be observed by the over-activation of receptor tyrosine kinases [83,87], indicating that *Drosophila* wrapping glia have the capability to wrap an axon with multiple membrane layers. The mutations in *sws* develop axonal and glial pathology and neuronal apoptosis. Additionally, the SWS protein was recently shown to play a role in the maintenance of neuromuscular function development and microtubule networks [88]. The *sws* gene is the ortholog of the neuropathy target esterase (*NTE*) gene, which is one of the genetic factors responsible for the development of hereditary spastic paraplegia [88]. In both vertebrates and invertebrates, the insulating of axons by glial processes is the only way to increase the conduction velocity of electrical impulses. Therefore, defects in this glial function can cause neurodegenerative diseases [89].

#### 3.2.4. Metabolic Support of Neurons

Glial cells in the nervous system play a critical role in the trophic support of neurons; thus, a lack of glial trophic support may result in the apoptotic death of neurons [90]. In mammals, the brain’s energy source is circulating blood glucose that is taken up by glial cells and converted to lactate [91,92,93]. The lactate is then transported from glia to neighboring neurons by membrane carriers which are known as monocarboxylate transporters (MCTs) [94,95,96,97,98].

In mammals, glia–neuron metabolic couplings between myelinating glia and axons, as well as between astrocytes and synapses, have been intensively studied. In the astrocyte–neuron lactate shuttle model, astrocyte-derived lactate acts as an energy substrate for neurons, while the uptake of glutamate released from synapses triggers the glycolytic production of lactate in astrocytes [99,100,101]. This concept has been expanded to other glial cells; it has been shown that oligodendrocytes uptake glucose in response to axonal glutamate release, thus controlling the metabolic cooperation between oligodendrocytes and axons far away from their neuronal cell body [102].

The *Drosophila* brain is rather small, so diffusion forces might be sufficient to account for the metabolic supply of nutrients to the brain region. However, the evolutionary conservation of the shuttle process of the energy supply has been demonstrated, and the metabolic coupling found in mammals has been shown to exist in *Drosophila*. The major energy metabolite in *Drosophila* is trehalose; the perineurial glial cells of *Drosophila* facilitate the absorption of this from the hemolymph. Absorbed trehalose is hydrolyzed into glucose by trehalase, which is present in the perineurial glia. It is then further metabolized to alanine or lactate, which can be taken up by neighboring neurons [94]. Since the gene expression of trehalase in neurons is very low in comparison to its high expression in the perineurial glia, neurons rely on glia for energy sources. It has been reported that the glial-specific knockdown of glycolysis results in neurodegeneration, while neuronal glycolysis is generally unnecessary [94]. Recently, the presence of a functionally active MCT was identified in the *Drosophila* CNS, implying its role in intercellular lactate shuttling in the *Drosophila* brain [103].

In addition to its role in the metabolic support of neurons, the glia–neuron lactate shuttle has been reported to play an additional role in neuroprotection under stress. Elevated reactive oxygen species (ROS) induce the formation of lipids in neurons that are stored as lipid droplets (LD) in glia [104]. Liu et al. (2017) showed that the lactate shuttle of *Drosophila* promotes neuronal lipogenesis in response to elevated ROS and LD accumulation in glia [105]. Impaired lipid transport or LD formation leads to neuronal degeneration and loss in both flies and neuron–glia co-cultured mouse primary cells. This suggests that LD accumulation in glia protects against neurodegeneration by scavenging peroxidated lipids.

#### 3.2.5. Glial Function in Neurotransmitter Recycling

Neuronal synapses are dynamic structures undergoing rapid formation and elimination, during which neurotransmitters are released. Glial cells are not only able to clear neurotransmitters from synaptic clefts, but they also play essential roles in the establishment and maintenance of proper synaptic connectivity by controlling structural and functional synaptic formation and by eliminating synapses [106]. The details are described below.

In mammalian brains, highly ramified astrocytes ensheath neuronal synapses with their fine processes (perisynaptic astrocyte processes) and are reported to be key players in synaptic development. Comparative analyses of *Drosophila* and mammalian astrocytes have revealed many conserved molecular signatures between these cell types [107,108] and, as in mammals, *Drosophila* astrocytes that surround neuronal cell bodies and proximal neurites are closely associated to synapses [109].

Astrocytes are linked to synapses in a functional as well as structural manner. Firstly, they possess the ability to recycle neurotransmitters. Astrocytes express several neurotransmitter transporters, such as the excitatory amino acid transporter 1 (EAAT1) and the γ-aminobutyric acid (GABA) transporter (GAT). These transporters work to break down excess synaptic glutamate or GABA, respectively, for the balance of excitatory/inhibitory synapses [38,61,110]. The astrocytic GAT activity in *Drosophila* is precisely regulated during synaptogenesis and behaves in an activity-dependent manner; it demonstrates neuron–glia crosstalk, the loss of which can have profound effects on the behavior and survival of neurons [39]. In addition to these neurotransmitter transporters, *Drosophila* astrocytic cells express glutamine synthetase and GABA transaminase [61,111]. These enzymes play a role in the metabolic breakdown of the transported glutamate and GABA into glutamine; this is then cycled back to neurons to provide the building blocks for ongoing synaptic transmission [112]. Recently, it was reported that the genes involved in glutamate and GABA cycling in *Drosophila* are transcriptionally regulated in glia to control their behavioral output, suggesting the importance of the transcriptional regulation of astrocytic neurotransmitter recycling in human disease [113]. In addition to astrocytic glial cells, it has been suggested that ensheathing glial cells also participate in the regulation of glutamate homeostasis [114].

Secondly, astrocytes display intracellular Ca^2+^ elevations in response to synaptic activity and modulate neuronal excitability [115]. Astrocyte-like glial cells in *Drosophila* are electrically coupled, as are their mammalian counterparts. Thus, the intracellular Ca^2+^ changes detected when activated by neurotransmitters show that a Ca^2+^ rise in astrocyte-like glial cells can trigger the rapid endocytosis of GAT from the astrocytic plasma membrane. This possibly induces opposing effects on neuronal excitability by increasing synaptic GABA levels that contribute to neuronal silencing [116,117,118].

Thirdly, the synaptic activity-driven astrocytic Ca^2+^ rise triggers the secretion of neuroactive molecules such as glutamate, D-serine, and ATP (gliotransmitters) from astrocytes via vesicular exocytosis [119]. The released gliotransmitters bind to presynaptic and postsynaptic receptors to regulate synaptic transmission, the disruption of which results in neuronal dysfunction and abnormal behaviors in animal models [120,121]. The use of genetic techniques has revealed that *Drosophila* astrocytic glial cells are also critical for the neural circuits regulating circadian behavior and sleep [122,123,124]. D-serine and its neuronal receptor have been implicated in sleep regulation in both flies and mammals [125,126,127]. Although there are numerous pieces of evidence regarding the roles of fly glial cells in circadian rhythmicity, little is known about *Drosophila* gliotransmitters in brains at present [128].

#### 3.2.6. Immunological Responses to Neural Brain Trauma

In *Drosophila* and vertebrates, glia play important roles in clearing degenerating neuronal projections, synapses, and apoptotic neurons during development, as described in Section 3.2.1. In addition to these clearing roles during development, glial cells appear to respond immunologically to nerve tissue damage in the brain by engulfing apoptotic debris [129]. In mammals, activated microglia clear up apoptotic cells and debris resulting from the injury. In contrast, in *Drosophila* adult brains, ensheathing glia primarily respond to injury by extending their membranes to injury sites and engulfing degenerating axons after acute nerve axotomy [37,130]. Musashe et al. (2016) found that the insulin-like signaling pathway and the downstream effector, Akt1, are activated in ensheathing glia after axotomy [131]. This activation is required for the upregulation of Draper via activated STAT92E, which subsequently induces the glial clearance of degenerating axonal debris [131]. In *Drosophila* larvae, it has also been shown that injury promotes the proliferation of Prospero-expressing astrocyte-like glial cells [132].

## 4. *Drosophila* Models of Human CNS Disorders Associated with Glial Dysfunction

In this section, we categorize CNS disorders into two groups: neurodevelopmental and neurodegenerative. In addition, we describe human CNS diseases that are affected by glial dysfunction, alongside their *Drosophila* models (Table 1). The potential roles of glial dysfunction in the onset and progression of these CNS diseases, along with recent insights into the molecular mechanisms of glial dysfunction, revealed using *Drosophila* models, are also described.

### 4.1. Models for Neurodevelopmental Disorders Associated with Glial Dysfunction

#### 4.1.1. Alexander Disease

Alexander disease (ALXDRD; OMIM #203450) is a rare neurological disorder caused by the dysfunction of astrocytes, in contrast to other neurodegenerative diseases which are induced by abnormalities in neurons. It is caused by heterozygous mutations in the glial fibrillary acidic protein (*GFAP*), which is an intermediate filament protein of astrocytes. The neuropathogenic manifestation of ALXDRD is astrocytic protein aggregates, termed Rosenthal fibers [163,164]. Patients with ALXDRD are classically characterized by different clinical features according to age. Recently, they were divided into two categories: type I (early onset) and type II (late onset). The phenotype of type I includes seizures, macrocephaly, motor delay, encephalopathy, failure to thrive, and typical brain magnetic resonance imaging (MRI) characteristics. The phenotype of type II includes autonomic dysfunction, ocular movement abnormalities, bulbar symptoms, and atypical MRI characteristics [165].

The molecular mechanisms of ALXDRD have been well characterized through studies using models ranging from human cell lines to *Drosophila*. Wang et al. (2011) created a *Drosophila* disease model that overexpressed the GFAP R79H mutant protein in glial cells [133]. The GFAP R79H mutant line shows similar clinical features to ALXDRD patients, including brain cell death, seizures, and protein aggregation in the form of Rosenthal fibers. The mutants also display JNK pathway and autophagy activation. As in the mouse model, the overexpression of other Rosenthal fiber components removes misfolded protein functions and alleviates GFAP toxicity in the *Drosophila* ALXDRD model. The neuronal cell death caused by GFAP toxicity is mediated by the glial glutamate transporter EAAT1 [133], while astrocytic nitric oxide functions as a novel signaling molecule that induces neuronal cell death through cGMP signaling. In addition, glial cell death occurs autonomously via DNA damage and p53, which is also observed in ALXDRD patient cell lines [134].

#### 4.1.2. Episodic Ataxia Type 6

Episodic ataxia (EA) is a neurological condition characterized by imbalance and the inability to coordinate muscular movements; it is often associated with progressive ataxia. Eight subtypes of EA have been identified, all caused by different genes [166,167]. EA6 (OMIM #612656) is caused by heterozygous mutations in the solute carrier family 1 member 3 gene (*SLC1A3*). The *SLC1A3* gene encodes a glutamate transporter that plays a role in the termination of excitatory neurotransmission in the CNS. SLC1A3, also known as EAAT1 or GLAST, is expressed in caudal brain regions such as the brainstem and the Bergmann astrocytes of the cerebellum. A proline-to-arginine (P > R) substitution, EAAT1 P290R, leads to decreased glutamate uptake in patients [168,169].

In *Drosophila*, dEAAT1 is expressed by astrocytes and other glia and functions as an anion channel. Interestingly, the expression of a mutant dEAAT1 P > R (EAAT1^P243R^) substitution protein in *Drosophila* glia induces defects in mature morphology and a decrease in filtration to the CNS neuropil. It also leads to the episodic paralysis of third-instar larvae, similar to the EA6 phenotype. In addition, the null mutant of EAAT1 cannot perform rhythmic peristaltic contractions. However, larvae that overexpress EAAT1^P243R^ reach the same maximum speed, suggesting that the neuronal circuits for locomotion develop normally. Expression of the Na^+^ -K^+^ -Cl^−^ co-transporter Ncc69, which normally allows chloride into cells, rescues the pathological effects caused by the expression of EAAT1^P243R^ [136,137]. Therefore, *Drosophila* disease model studies suggest that the clinical features of EA6 are caused by abnormal ion exchange and *EAAT1* mutations in astrocytes.

#### 4.1.3. Fragile X Syndrome

Fragile X syndrome (FXS; OMIM #300624) is the most common X-linked disease of inherited intellectual and emotional disability, which is the cause of autism disorder [170]. Patients with FXS show cognitive impairment, autistic features, attention deficits, increased rates of epilepsy, and motor abnormalities [171]. More than 200 trinucleotide repeats (CGG) in the 5′ untranslated region (UTR) of the fragile X mental retardation 1 gene (*FMR1*, also known as the gene for FMRP translational regulator 1) are responsible for FXS. The CGG repeats in the 5′ UTR induce DNA methylation of *FMR1*, which leads to transcriptional silencing [172]. The absence of the FMR1 protein (FMRP), an RNA binding protein that regulates protein synthesis in neuronal dendrites, leads to the upregulation of target mRNAs, subsequently inducing defects in synaptic strength and immature dendritic spines [173]: in mice, *Fmr1*-knockouts have a phenotype that presents many of the clinical and cell-biological features of the disease, including abnormal dendritic spine development. Hippocampal neurons grown on *Fmr1*-deficient astrocytes show abnormal dendritic morphology relative to those grown on wild-type astrocytes [173]. Furthermore, the intrinsic dendritic defects of *Fmr1*-deficient neurons are significantly rescued when these cells are grown on a monolayer of wild-type astrocytes rather than *Fmr1*-deficient astrocytes. *Fmf1*-deficient astrocytes therefore appear to have an effect on synapse formation, at least in culture [174].

In humans, there are two paralogs of the *FMR1* gene: *FXR1* and *FXR2*. However, there is only one ortholog of *FMR1* in *Drosophila*: *dFMR1* [175]. The dFMR1 protein is expressed in the CNS, testis, and eye disc in larvae [176]. In adults, it is expressed in brain neuronal cell bodies but not glia. Null mutant and overexpression lines of *dFMR1* show defects in synaptic formation and synaptic transmission via Futsch, a microtubule-associated protein homologous to mammalian MAP1B. Both lines of *dFMR1* mutants have phenotypes, including eclosion and neural structure, and behavioral defects, including behavior related to circadian rhythm, locomotion, sleep, and learning [177,178]. A recent study revealed that FMRP is related to neuronal stem cell proliferation and differentiation. Interestingly, during early brain development, FMRP is essential for regulating the exit from the quiescence state in neuroblasts as well as glial cells. Loss of FMRP in the glial lineage leads to an increase in cyclin E positive cells and activation of the insulin pathway of neuroblasts [138]. Therefore, neuronal and glial cells may contribute to the phenotypes in *FMR1* mutants in both a non-cell-autonomous and cell-autonomous manner. This raises the possibility that the in vivo defects in neuronal synaptic formation are at least partly related to neuron–glia interactions during development.

#### 4.1.4. Rett Syndrome

Rett syndrome (RETT; OMIM #312750) is an X-linked progressive neurodevelopmental disorder which occurs exclusively in females. Patients with RETT develop normally up to 18 months of age. Thereafter, they gradually develop autism, stereotypic hand wringing, respiratory abnormalities, microcephaly, seizures, and mental retardation [179,180]. RETT is caused by mutations in the gene for methyl-CpG-binding protein 2 (*MECP2*). MECP2 is a well-known transcriptional repressor that inhibits gene expression via two epigenetic markers: DNA methylation and histone acetylation. MeCP2 domains include a methyl-CpG-binding domain and a transcriptional repression domain, which interact with histone deacetylase and mSin3a [181,182]. Mice with loss-of-function mutations in *Mecp2* display many features similar to RETT [180,183,184]. Although RETT is generally attributed to neuronal dysfunction, *Mecp2*-null astrocytes are unable to support the normal dendritic ramification of neurons in culture [185]. Furthermore, *Mecp2*-deficient mouse microglial cells release an abnormally high level of glutamate that may cause synaptic abnormalities [186], suggesting a pathophysiological role for glial MCP2 in neurological diseases.

In *Drosophila,* no *MECP2* ortholog exists. Thus, numerous studies have been conducted using the gain-of-function mutant of human *MECP2*. In *Drosophila,* MECP2 is phosphorylated at serine 423 and interacts with sin3a, N-coR, and REST. Overexpression of *MECP2* in *Drosophila* induces abnormal developmental phenotypes such as disorganized eyes and ectopic wing veins. In addition, the overexpression of *MECP2* in neurons induces defects in dendrite structure and motor behavior, although the electrophysiological properties of neurons are not affected. This indicates that MECP2 has a specific and important role in dendrite formation [187,188]. A recent study revealed that MECP2 functions in glial cells as well as neurons. Interestingly, astrocyte-specific *MECP2* overexpression in *Drosophila* causes a reduction in the total time spent asleep and induces deficits in sleep maintenance. Defects in sleep behavior are neuropsychological features of RETT, suggesting that abnormal sleep patterns might be caused by *MECP2* overexpression in astrocytes [139]. Through *Drosophila* disease models, it has been revealed that the ectopic expression of *MECP2* induces deficits in neuronal development and behavior, which is regulated by glial cells, especially astrocytes.

#### 4.1.5. Schizophrenia

Schizophrenia (SCZD) is a severe psychiatric disorder with a prevalence of approximately 1% in studied populations. It is associated with multiple gene loci in conjunction with epigenetic, stochastic, and environmental factors. SCZD is characterized by hallucinations, delusions, thought disorders, and blunted emotions [189].

Schizophrenia appears to be highly heritable, but the genetics are complex [190]. There may not be a single cause; numerous SCZD susceptibility genes have been identified thus far. Straub et al. (2002) were the first to report that single nucleotide polymorphisms within the gene for dystrobrevin-binding-protein 1 (*DTNBP1*), also known as dysbindin (*dysb*), were strongly associated with SCZD [191]. This gene is currently considered one of the candidate genes for SCZD3 (OMIM #600511). DTNBP1 is a component of the biogenesis of lysosome-related organelles complex 1 and plays a role in glutamate neurotransmission. It is widely expressed in the brain [192,193], and a role of glial DTNBP1 expression in SCZD has been suggested in a *Drosophila* model. A mutant of the *Drosophila dysb* gene (*Ddysb*) leads to the abnormal modulation of neurotransmission and calcium influx. Furthermore, the null mutant of *Ddysb* impairs memory, locomotion ability, and mating orientation. Interestingly, impaired memory is rescued only by expressing *Ddysb* in neuronal cells, whereas abnormal locomotion activity and mating orientation are restored only by expressing *Ddysb* in glial cells. In addition, expression levels of the dopamine metabolic enzyme, ebony, in glial cells are reduced in the *Ddysb* mutant. This indicates that *Ddysb* regulates dopamine metabolism in glial cells [140]. Taken together, these results indicate that *Ddysb* mutations cause different functional defects in glial cells than in neuronal cells. Therefore, defects in *DTNBP1* may differentially contribute to the phenotypes of SCZD depending on the affected cell types.

#### 4.1.6. Sotos Syndrome 1

Sotos syndrome 1 (SOTOS1; OMIM #117550) is a developmental disorder characterized by facial abnormalities, advanced bone age, and macrocephaly. Patients with SOTOS1 also have neurological defects such as mental retardation, seizures, and delayed language and motor development [194,195,196]. SOTOS1 is caused by haploinsufficiency of the gene for nuclear receptor-binding SET domain protein 1 (*NSD1*). The encoded protein, NSD1, is a histone methyltransferase, which catalyzes the mono- and di-methylation of histone 3 at lysine 36 (H3K36) [197,198]. Interestingly, patients with *NSD1* duplication have reciprocal phenotypes such as microcephaly and developmental delay, indicating that NSD1 is highly associated with brain development [199].

The *Drosophila NSD1* ortholog, *NSD*, interacts with various chromatin remodeling factors, such as HP1 and insulator binding protein dCTCF/Beaf32, acting downstream of the DRE/DREF pathway [200,201,202]. Overexpression of *NSD* in *Drosophila* recapitulates the phenotypes of human *NSD1* duplication disease, such as developmental delay, and induces apoptosis through a JNK signaling pathway [203]. Interestingly, *NSD* overexpression in glial cells, not neurons, induces brain cell death, behavioral defects, and a decrease in brain size that is similar to the microcephaly phenotype of patients with *NSD1* duplication. Among glial subtypes, the dysfunction of astrocyte-like glia appears to be the main cause of the *NSD* overexpression phenotypes that accompany learning disabilities and abnormal circadian rhythms [141]. These results from *Drosophila* suggest that the human *NSD1* duplication disease may be caused by astrocyte dysfunction. However, the molecular mechanisms by which NSD1 plays a role in human *NSD1* duplication disease occurrence, as well as in SOTOS1, remains elusive.

### 4.2. Models for Neurodegenerative Diseases with Abnormal Glial Function

#### 4.2.1. Alzheimer’s Disease

Alzheimer’s disease (AD) is the most common neurodegenerative disorder worldwide, affecting 60% to 80% of all people with dementia [204]. The affected brain exhibits astroglyosis and neuronal loss, and is characterized by two features: the extracellular accumulation of amyloid beta (Aβ) in senile plaques and the intracellular existence of neurofibrillary tangles composed of hyperphosphorylated tau filaments [205]. AD develops due to multiple factors rather than a single cause, and researchers have found several genes that increase the risk of AD. Although the relationship between these genes and AD characteristics remains unclear, four different familial AD subtypes caused by genetic mutations, AD1 to AD4, have been identified. Here, we summarize two AD subtypes, AD1 and AD2, which are well modeled in *Drosophila* and are associated with glial dysfunction.

AD1 (OMIM #104300) is one of the autosomal dominant forms of early onset AD, which is caused by mutations in the gene encoding the amyloid precursor protein (*APP*) [206]. APP is an integral membrane protein that is expressed ubiquitously, with essential roles in neuronal signaling, intracellular transport, and neuronal homeostasis [207]. APP undergoes post-translational proteolysis/processing to generate insoluble Aβ peptides [208]. Familial mutations in the *APP* gene elevate the production of Aβ in early onset AD; Aβ deposition in the brain can lead to synaptic dysfunction, neuronal cell death, impaired learning and memory, and abnormal behaviors [209].

*APP* is conserved across various species and its ortholog can be found in invertebrates including *Drosophila* [210]. In *Drosophila*, however, some components of APP proteolytic processing are not conserved and the *Drosophila APP* ortholog, dAPP-like (*dAPPL*), does not contain the Aβ domain. It is subsequently unable to be cleaved in the fly [211]. Nonetheless, *Drosophila* has been used as an excellent model organism for AD. Both deletion mutants and flies overexpressing either *dAPPL* or human *APP* exhibit defects in axonal transport (albeit Aβ-independent) [212]. Furthermore, the ectopic expression of human Aβ42 peptides in the fly brain causes age-dependent learning defects and short-term memory impairment due to their elevated aggregation propensity. In addition to Aβ42, several other Aβ peptides that display different levels of toxicity as well as aggregation have been reported [213]. Jonson et al. (2015) demonstrated that the neuronal expression of Aβ42 peptides that are mutated at the N- or C-terminal differentially contribute to Aβ toxicity [213]. N-terminal truncated peptides present a less severe phenotype but are quite toxic, while mutating the C-terminal residue 42 in Aβ42 greatly reduces Aβ accumulation and toxicity.

In addition to numerous studies on the role of neuronal APP and Aβ, their functions in glial cells have also been explored. The knockdown of *dAPPL* in adult fly glia alters sleep patterns, possibly disrupting the glial dAPPL role in glutamate metabolism, recycling, or both [142]. This is intriguing due to the mutual relationship between AD and sleep disruption [214]. As for Aβ peptides, their neurotoxicity can be reduced by the glial expression of the engulfment receptor Draper, which reverses locomotor defects and extends lifespan [143]. Similarly, glial expression of the connective tissue growth factor (CTGF) reduces Aβ deposits and improves both locomotor function and memory deficits in *Drosophila*; this is also shown in primary mammalian glial cells [144]. This result is noteworthy as it highlights the evolutionarily conserved roles between fly and mammalian glial cells in AD progression.

AD2 (OMIM #104310), also known as late onset AD2 or AD associated with APOE4, is associated with *APOE*. The encoded protein APOE is primarily produced in astrocytes and microglia and plays a major role in glia–neuronal lipid metabolism. There are three alleles of the *APOE* gene, E2, E3, and E4; of these, the E4 variant significantly increases the risk of late onset AD [215]. No fly gene has been identified as orthologous to human *APOE*. In a human *APOE* transgenic *Drosophila* model, the E4 variant exhibits a defective response to oxidative stress, resulting in increased neurotoxicity, whereas the E3 variant has a neuroprotective effect [216]. APOE4 expression in glial cells inhibits the neuroprotective formation of LDs, leading to neuronal cell death [145]. Interestingly, the glial knockout of *GLaz*, the *Drosophila APOD* ortholog, is rescued by the glial expression of human APOE2 and 3, but not APOE4. This indicates that human APOE can be functionally substituted for the fly glial apolipoprotein and that APOE4, the AD risk allele, is impaired in lipid transport and promotes neurodegeneration [145]. This fly model provides insights into glial roles in lipid metabolism in AD progression.

#### 4.2.2. Amyotrophic Lateral Sclerosis 1

Amyotrophic lateral sclerosis (ALS) is a genetically heterogeneous disorder with several causative genes, and it is a neurodegenerative disorder characterized by the death of motor neurons in the brain, brainstem, and spinal cord, resulting in fatal paralysis. ALS1 (OMIM #105400), also known as familial ALS (FALS), comprises approximately 10% of all ALS cases [217]. Mutations in the Cu/Zn superoxide dismutase 1 gene (*SOD1*) account for approximately 20% of ALS1 and 2% of all ALS cases [218]. SOD1 is a ubiquitously expressed and highly conserved antioxidant enzyme that catalyzes the dismutation of superoxide into oxygen and hydrogen peroxide [218].

SOD1-linked ALS1 is thought to function in a non-cell-autonomous manner such that motoneurons are critical for the onset, while glia contribute to the progression of the disease [219]. Glial roles in neuronal support are also being increasingly demonstrated in ALS1 animal models. The most widely used mouse model of ALS1 is based on the expression of the human mutant *SOD1* gene from ALS1 patients. *SOD1* mice exhibit a phenotype similar to ALS1 due to the expression of the mutant gene in specific glial cells such as microglia, astrocytes, and oligodendrocytes [220,221,222]. Interestingly, the neuron-specific expression of mutant *SOD1* in mice is sufficient to trigger neuronal degeneration cell-autonomously as well, with dramatic wild-type *SOD1* aggregation in oligodendroglia after the onset of neuronal degeneration [223].

Several attempts have been made to model *SOD1*-linked ALS1 in flies. Similar to the mouse model, expression of the mutant protein with human disease-related *SOD1* alleles results in the dysfunction of both neuronal and glial cells in *Drosophila*. The cell-specific expression of human *SOD1* mutants in either type decreases motor function, lifespan, sensitivity to hydrogen peroxide, and lipid peroxidation [146], providing evidence for the non-cell-autonomous roles of the two cell types in the onset and progression of ALS1.

#### 4.2.3. Amyotrophic Lateral Sclerosis 10 with or without Frontotemporal Dementia

Amyotrophic lateral sclerosis 10 (ALS10; OMIM #612069), also known as frontotemporal lobar degeneration with TDP43 inclusions, TARDBP-related (FTLD-TDP, TARDBP-RELATED), is caused by heterozygous mutations in the TAR DNA binding protein gene (*TARDBP*), which encodes the TDP-43 protein. TDP-43 is abundantly expressed in nearly all tissues and is a well-conserved protein, including in *Drosophila* [224]. The role of normal TDP-43 is the regulation of RNA, including mRNA splicing, stability, translation, and transcription. The protein therefore associates with members of the heterogeneous nuclear ribonucleoprotein family [225]. TDP-43 aggregation and neuropathology are observed in ALS motor neurons, suggesting a central role for TDP-43 in neurodegenerative disease pathogenesis [226].

Similar to several genes implicated in neurodegeneration, both loss- and gain-of-function have been proposed to mediate TDP-43 pathogenesis. Loss of murine *TDP-43* causes early embryonic lethality and disrupts motor function [227]. In transgenic mice expressing a mutant human TDP-43 protein, typical features of ALS, such as progressive gait abnormalities and short lifespan, are induced, possibly due to the accumulation of ubiquitinated proteins that are associated with neuronal loss and increased glial reaction [228].

*Drosophila* has an endogenous ortholog of human TDP-43, a TAR DNA-binding protein-43 homolog protein (TBPH). Since TBPH can functionally substitute for human TDP-43 in many assays, ALS can be modeled in the *Drosophila* system to investigate the function of TDP-43 in both normal and disease conditions [229]. The contribution of glial cells to ALS10 pathology has also been reported in the *Drosophila* model [147,148]. Similar to its human ortholog, both loss and gain of *TBPH* functionality in either muscle or glial cells can lead to cytological and behavioral phenotypes of ALS10 in *Drosophila*. Using this model, the glutamate transporters, dEAAT1 and dEAAT2, were identified as potential direct targets of TBPH function [147]. Relationships between astrocytic dysfunction and glutamate transport have been well documented in ALS patients; astrocytes from the brains of ALS patients exhibit selective decreases in EAAT-2, the main glutamate transporter in the CNS [230,231,232]. Thus, these *Drosophila* data support the association of TDP-43 with glial dysfunction in ALS10 neurodegeneration in a non-cell as well as a cell-autonomous manner [148].

#### 4.2.4. Ataxia Telangiectasia

Ataxia telangiectasia (AT; OMIM #208900) is an autosomal recessive disorder characterized by cerebellar ataxia, telangiectasia, and predisposition to malignancy and immune disorders [233]. AT is caused by mutations in the ataxia telangiectasia mutated gene (*ATM*), coding for the serine/threonine protein kinase, ATM. ATM is known to be a master regulator of the DNA double-strand break repair pathway and patients with loss of ATM functionality display extreme sensitivity to DNA damage [234]. ATM also plays a conserved role in telomere maintenance [235]; the list of its functions is steadily increasing and includes roles in oxidative stress and autophagy [236].

The *ATM* gene is conserved from fungi to humans [237]. The gene encoding the *Drosophila* ATM kinase, *dATM*, was originally designated *tefu* due to the telomere fusion defects observed in mutant flies [238]. The characteristic biological and molecular functions of ATM are conserved in *Drosophila*; it therefore provides a useful animal model for analyzing the molecular functions of ATM. For example, conditional loss-of-function *dATM* alleles in flies cause progressive neurodegeneration through the activation of the innate immune response [239]. Unlike in mammals, although the *ATM* null mutant is lethal during fly development, the identification of conserved regions in the *Drosophila* ATM protein is expected to facilitate the understanding of conserved ATM mechanisms. These are important for telomere maintenance, DNA repair, and neurodegeneration [240,241]. The contribution of glial cells to AT pathology has also been reported. In an AT fly model, a specific *ATM* knockdown in glial cells activates the innate immune response transcription factor NF-κB homolog, Relish, as a primary driving force for the degeneration of neurons and glia, resulting in reduced mobility and decreased lifespan [149,150]. This suggests that manipulating glial immunity pathways could serve as a useful strategy to extend lifespan.

#### 4.2.5. Friedreich’s Ataxia

Friedreich’s ataxia (FA; OMIM #229300), also known as FRDA1, is a progressive neurodegenerative disorder and a subtype of autosomal recessive cerebellar ataxia characterized by progressive gait and limb ataxia [242]. FA can shorten life expectancy; death arises most frequently from cardiomyopathy and cardiac failure rather than from neurological effects [243]. It is caused by the expansion of 70 to over 1000 GAA triplets in the first intron of the frataxin gene (*FXN*). This inhibits the mRNA expression of FXN by gene silencing via epigenetic changes [244,245,246].

FXN is a highly evolutionarily conserved mitochondrial protein [247]. The cellular function of FXN is critical for life in multicellular organisms in that its strong reduction in *Drosophila* seriously affects viability, while knockout in mice is embryonic lethal [248,249,250]. Although the function of FXN is not clear, the protein is involved in the assembly of iron–sulfur clusters, which are important for optimal mitochondrial function [251]. Lack of *FXN* causes an energy synthesis reduction in the mitochondria and extra ROS generation through excess iron, leading to further cell damage [252]. In developing mice, *FXN* depletion in neuronal and astrocyte precursors results in severe ataxia and early death, associated with growth and survival impairments in cerebellar astrocytes. In addition, abnormal levels of antioxidant enzymes are induced in the cerebellum, suggesting a role for astrocytic oxidative stress in FA neurodegeneration [253].

Similarly, the importance of glial dysfunction in the pathogenesis of FA has been demonstrated in *Drosophila*. The glial deletion of *fh*, the *Drosophila FXN* ortholog, induces an increase in fatty acids that catalyzes an enhancement of lipid peroxidation levels that can be visualized by the accumulation of LDs in glial cells. This FA model fly shows FA-like symptoms such as a reduced life span, an increased sensitivity to oxidative insult, neurodegeneration, and impaired locomotor performance, which can be explained by an increase in intracellular toxicity through lipid peroxides. In this fly FA model, glial expression of the *Drosophila* apolipoprotein D (*APOD*) ortholog, *GLaz*, shows a strong protective effect by lowering lipid peroxide levels, subsequently reducing glial cell death [151]. Furthermore, *FXN*-depleted cultured human astrocytes undergo cell death due to mitochondrial dysfunction and high oxidative stress. This suggests that astrocytic cell death by oxidative stress may cause the neuronal toxicity seen in FA patients [254]. In addition to mitochondrial dysfunction, endoplasmic reticulum (ER) stress was recently reported to be a novel and crucial player in the progression and etiology of FA. In a *Drosophila* FA model, the downregulation of Marf, a protein known to play crucial roles in the interface between ER and mitochondria, in glial cells improves motor function, neuronal degeneration, and lipid homeostasis through the suppression of ER stress [152]. Very similar results were obtained through ER stress reduction by treatment with small chemical compounds known to decrease ER stress.

#### 4.2.6. Frontotemporal Dementia and/or Amyotrophic Lateral Sclerosis 1

Initially, frontotemporal dementia (FTD) and ALS, two major neurodegenerative conditions, were classified as two separate disorders due to their clinically distinct phenotypes: pure cognitive impairment (FTD) and pure movement impairment (ALS). However, up to 50% of all patients present some degree of both ALS and FTD phenotypes, suggesting a continuous FTD and ALS spectrum, termed FTD and/or ALS (FTDALS).

FTDALS is an autosomal dominant neurodegenerative disorder characterized by adult onset. FTDALS is genetically and pathologically heterogeneous [255], with more than 100 genes currently known to contribute to this disorder [256]. Among the various FTDALS genes, the chromosome 9 open reading frame 72 gene (*C9ORF72*) has been identified as the major causative gene of familial forms of FTDALS, termed FTDALS1 (OMIM #105550). FTDALS1 is caused by a heterozygous hexanucleotide repeat expansion (HRE; GGGGCC) in a noncoding region of the *C9ORF72* gene, ranging in size from 250 to more than 2000 repeats. C9ORF72 is a guanine nucleotide exchange factor that may function as a dual exchange factor, coupling physiological functions such as cytoskeleton modulation and autophagy with endocytosis [257]. Its HRE elicits a complex cascade of events due to the deregulated functions that contribute to neurodegeneration. HRE in *C9ORF72* can induce the formation of TDP-43 inclusions; therefore, the presence of TDP-43 inclusions characterizes patients with FTDALS1 [258,259]. Recently, this repeat expansion in *C9ORF72* was also shown to influence immune dysregulation, subsequently mitigating neuroinflammatory disease processes in ALS [260]. In studies using FTDALS1 patient-derived cells, *C9ORF72* HRE was shown to reduce *C9ORF72* expression, causing neurodegeneration via the accumulation of glutamate receptors along with an impaired clearance of neurotoxic dipeptide repeat (DPR) proteins [261]. In addition, using human iPSC-derived astrocytes, an astrocytic *C9ORF72* mutation was confirmed to be responsible for both cell-autonomous astrocyte pathology and non-cell-autonomous motor neuron pathophysiology [262].

*C9ORF72* mRNA and its protein are highly expressed in normal mice and human brains; loss-of-function homozygous mutations in *C9orf72* lead to premature death in mice [263,264]. However, the neural-specific ablation of *C9orf72* in the conditional knockout mice is insufficient to cause motor neuron disease [265]. Despite a recent report showing that reduced *C9orf72* function increases C9orf72 HRE toxicity [266], the gain-of-function of *C9orf72* HRE has been known to play more crucial roles in neurodegenerative changes. For this, two molecular mechanisms have been suggested: toxicity from HRE-containing RNA and the accumulation of toxic DPR proteins via RAN translation [267].

Compared to the crucial roles of neurons in motor and behavioral phenotypes in FTDALS1 identified by mouse models expressing C9orf72 DPR proteins [268,269], few studies have demonstrated a direct glial contribution to FTDALS1 neuropathology [261]. Recently, DPR proteins in transgenic mice were shown to non-cell-autonomously trigger key features of FTDALS1, including cytoplasmic mislocalization and aggregation of TDP-43, suggesting a possible role of glial cells as the neighboring cells of degenerating neurons [270].

*C9ORF72* is highly conserved in evolution [271], but no *Drosophila* ortholog has been identified. However, *Drosophila* has been widely used to model *C9ORF72* HRE RNA and DPR protein toxicity through the generation of transgenic flies. In an FTDALS1 *Drosophila* model, the C9ORF72 DPR protein causes neurodegenerative phenotypes and excitotoxicity in glutamatergic neurons [272]. Using novel *Drosophila* FTDALS1 models that produce spliced intronic nuclear HRE RNA or poly(A)+ DPR mRNA exported to the cytoplasm for protein production, it was revealed that toxicity was correlated with HRE-derived DPR protein production but not HRE-RNA accumulation [153]. In addition, a vicious feedback loop was recently found in C9ORF72 DPR transgenic flies, in which DPR proteins, but not HRE RNA accumulation, regulate TDP-43 dysfunction and TDP-43 inversely increases levels of DPR proteins [273].

#### 4.2.7. Polyglutamine-Related Disorders

Polyglutamine (polyQ) diseases are dominantly inherited genetic disorders resulting from CAG repeat expansions within the respective disease genes [274]. The accumulation of insoluble aggregates of misfolded proteins with pathogenic polyQ tracts is a hallmark of late onset neurodegenerative diseases. Misfolded proteins deposited in neuronal cytoplasm or nuclear inclusion bodies have been detected in the brains of patients and transgenic mice [275]. These diseases are monogenic and are inherited dominantly. To date, nine polyQ diseases have been described: Huntington’s disease (HD), X-linked spinobulbar muscular atrophy (Kennedy disease), spinocerebellar ataxia (SCA) 1, SCA2, SCA3 (also known as Machado–Joseph disease), SCA6, SCA7, SCA17, and dentatorubral pallidoluysian atrophy [276]. Since the disease-linked proteins share no homology apart from the polyQ tract, the polyQ expansion may be the common pathogenic mechanism leading to these neurodegenerative diseases. Several of the polyQ proteins are expressed in glia, which appear to be involved directly in polyQ pathogenesis [277,278]. Associations between some of these proteins and abnormal glial function have been reported in the *Drosophila* models of HD, SCA1, SCA3, and SCA7. Since most of the disease-causing human genes have no *Drosophila* orthologs, especially pathogenic mutant polyQ expanded alleles, *Drosophila* models of polyQ diseases have been generated by introducing human pathogenic versions of these genes into flies.

##### Huntington’s Disease

HD (OMIM #143100) is an autosomal dominant disease that presents cognitive decline and motor symptoms, particularly chorea. HD is caused by the unstable expansion of CAG repeats within the coding region of the interesting transcript 15 (*IT15*, also known as huntingtin (*HTT*)) gene on 4p16.3. HD occurs when more than 37 polyQ repeats are present in the HTT protein [279].

The effect of expanded polyQ repeats on neurons has been well studied in *Drosophila*. Ectopic expression of exon 1 of human *IT15* containing 75 or 120 polyQ repeats in *Drosophila* causes late onset progressive neurodegeneration. This is dependent on repeat length and is a representative feature of HD [280]. The ability of HTT-containing human expanded repeats to induce cell death in fly neurons is noteworthy because it indicates the presence of a neurodegeneration process that is evolutionarily conserved. Interestingly, the knockdown of *htt*, the *Drosophila* ortholog of the *IT15* gene, causes axonal transport defects as well as an HD-like phenotype similar to that found with overexpression of the human gene [281], indicating the importance of its proper expression.

HTT is broadly expressed in the nervous system and alterations in its expression in glial or neuronal cells can contribute to the HD disease pathology [277,282]. Thus, neuronal dysfunction and death are caused by both neuron-intrinsic mechanisms and alterations in glia–neuron communication. An HD-like phenotype is induced by expressing a mutant HTT protein (mHTT) selectively in *Drosophila* glia [154]. Besson et al. (2010) demonstrated that glial pathology due to mHTT in HD flies is alleviated by increased energy metabolism, but neuronal pathology in HD flies is not rescued [155], suggesting that increasing glucose metabolism may be beneficial in rescuing abnormal glia–neuron communication in HD, since defects in energetic metabolism are involved in mHTT-induced glial alterations.

Interestingly, the glial scavenger receptor Draper may affect HD pathology via the uptake of neuronal mHTT aggregates into *Drosophila* phagocytic glia [156]. Pearce et al. (2015) showed that glia can clear the mHTT aggregates expressed in neurons via Draper and subsequent phagocytic engulfment [156]. However, surprisingly, the internalized mHTT aggregates via Draper promote the prion-like pathogenic conversion of soluble wild-type HTT within the glial cytoplasm without degradation within the membrane-enclosed phagolysosomal system. As numerous studies have demonstrated compelling evidence for glial phagocytosis having a neuroprotective role, this observation is surprising in that glia may potentially act as reservoirs of prion-like species in the progression of neurodegenerative diseases, possibly facilitating their spread to other cells. This implicates Draper/MEGF10 as a novel therapeutic candidate for blocking the spread of protein aggregates in specific neurodegenerative disorders.

##### Spinocerebellar Ataxia Type 1

Autosomal dominant cerebellar degenerative disorders are generally referred to as SCAs [283]. However, significant phenotypic overlap between different forms of SCAs and significant phenotypic variability within each subtype has been reported [284]. SCA type 1 (SCA1; OMIM #164400) is characterized by neurological symptoms including dysarthria, hypermetric saccades, and ataxia of gait and stance.

This cerebellar dysfunction is caused by expanded CAG trinucleotide repeats in the *ataxin-1* gene. Ataxin-1 is present in the nuclear region of various brain regions and in non-neuronal tissues, and it is thought to be involved in transcriptional repression [285,286]. The polyQ expansion in *ataxin-1* is variable in length, but 39 or more uninterrupted CAG triplets cause disease, while longer repeat tracts are correlated with earlier onset and faster progression [287,288]. Mutant ataxin-1 spontaneously misfolds and forms aggregates in cells [289]. Interestingly, its aggregation is not required for pathogenesis in mice, but its polyQ expansion is associated with impaired proteasomal degradation and is thought to affect interactions with other cellular proteins that regulate the expression of ataxin-1 target genes, leading to disease [290,291].

Ataxin-1 is conserved across species, including *Drosophila* [292]. In a *Drosophila* SCA1 model targeting the nervous system, the expression of a human mutant *ataxin-1* containing a polyQ expansion causes a late onset, progressive motor impairment phenotype [293]. In *Drosophila* HD and SCA1 models targeting glial cells, mutant *HTT* and *ataxin-1* genes, respectively, were associated with an accumulation of protein fragments in intranuclear inclusions inside cells. This was accompanied by the progressive degeneration of glial and neuronal cells, suggesting a non-cell-autonomous effect of glial cells [157]. In additional comparisons of fly models of the two diseases, differences in stage-specific toxicities were observed between the two polyQ-expanded proteins, HTT and ataxin-1. Ataxin-1 expansion induced severe developmental phenotypes when expressed in glial cells, whereas the developmental effect of HTT was generally weak, except when induced in cholinergic neurons [158]. Furthermore, when expressed by the pan-glial cell specific Repo drivers, the developmental effect of the expanded ataxin-1 on the earliest stages of glial development is more prominent than that at later stages. This suggests a possible role for glial polyQ expanded ataxin-1 pathology in the early onset of the disease.

##### Spinocerebellar Ataxia Type 3

SCA type 3 (SCA3; OMIM #109150), also known as Machado–Joseph disease, is an autosomal dominant progressive neurologic disorder that is characterized principally by ataxia that is similar to that of SCA 1 but with peak saccade velocity and presenting extrapyramidal signs more often. SCA3 is the most frequent among the SCA subtypes, comprising approximately 21% of the worldwide cases of autosomal dominant cerebellar ataxias [294]. This dominantly inherited disorder is linked to an unstable CAG repeat in the *ataxin-3* gene [295]. Ataxin-3 belongs to the family of cysteine proteases that exhibit deubiquitinase activity and was recently reported to play a role in the transcriptional regulation of multiple signaling pathways, suggesting its physiological function in normal cells [296,297]. Although *ataxin-3* is evolutionarily conserved, including in *Caenorhabditis elegans*, its ortholog has not been identified in *Drosophila* [298]. In transgenic *C. elegans* SCA3 models, the pan-neuronal expression of mutant ataxin-3 leads to a polyQ-length dependent, neuron subtype-specific aggregation and subsequent neuronal dysfunction [299], which are suppressed by the activation of serotonergic signaling. This suggests that the modulation of serotonergic signaling is a promising therapeutic target for SCA3 [300]. In SCA3 mouse models, pathogenic *ataxin-3* knock-in induces the neuronal accumulation and aggregation of mutant ataxin-3 but, interestingly, not a behavioral phenotype [301]. However, ataxin-3 neuronal aggregation was found to be primarily linked to modulated gene expression in oligodendrocytes, the myelinating glia of the CNS. This suggests that oligodendrocyte dysfunction is a primary toxic gain-of-function mechanism in SCA3 disease pathogenesis, as oligodendrocyte dysfunction could impede the propagation of neuronal action potentials [301]. In addition, a new humanized *ataxin-3* knock-in mouse model displaying late disease onset produces SCA3 neuropathology in both neurons and glia [302].

The *Drosophila* SCA3 model was generated by introducing the pathogenic version of *ataxin-3*. It presents with key features of the human disease, including neuronal degeneration, nuclear inclusions, and trinucleotide repeat instability, including large changes in repeat length and a strong expansion bias [303]. These phenotypes can be partially rescued by the co-expression of the molecular chaperone HSP 70 or by suppressing apoptosis and histone deacetylation [265,304]. The pathogenicity of the truncated ataxin-3 is more severe than that of the full-length protein, due at least in part to the protective nature of functional domains in the normal protein [305]. Several experiments have demonstrated the necessity of proper glial function for neuronal survival in SCA3 fly models. Although behavioral deficits are induced by the overexpression of ataxin-3 in both fly glia and neurons, only glia are susceptible to the toxic action of the protein, suggesting that neurological symptoms may be caused by glial dysfunction in a non-cell-autonomous manner [159]. In addition, the expression and aggregation of polyQ-expanded ataxin-3 in fly glia is deleterious in that specific expression in all glia or in the BBB glia causes cell-autonomous damage to the BBB’s integrity and reduces lifespan [160]. On the other hand, in response to the proteotoxic stress induced by polyQ expanded SCA3 expression or the amyloid beta peptides associated with AD, astrocytes in fly neurons activate Relish, subsequently enhancing neurodegeneration. It has been shown that specific Relish inhibition in astrocytes delays neurodegeneration and extends lifespan. These data provide further evidence for the non-cell-autonomous contributions of astrocytes to neurodegeneration, albeit from an opposing perspective [161].

##### Spinocerebellar Ataxia Type 7

SCA type 7 (SCA7; OMIM #164500) is an autosomal dominant disorder characterized by the adult onset of progressive cerebellar ataxia similar to that of SCA 1 but with retinal degeneration. It is caused by a heterozygous expanded CAG repeat in *ataxin-7*. Human ataxin-7 is a subunit of the SPT3/TAF9/GCN5 acetyltransferase complex, the mammalian Spt/Ada/Gcn5 acetylase (SAGA)-like complex, which appears to be critically important for deubiquitination and chromatin remodeling at the level of histone acetylation [306]. Recently, it was shown that endogenous ataxin-7 is modified by SUMO proteins and that the SUMO pathway contributes to the clearance of aggregated ataxin-7 in the cerebellum and retina of SCA7 knock-in mice. This suggests that its deregulation might be associated with SCA7 disease progression [307]. The effect of glial dysfunction on the promotion of neuronal dysfunction and degeneration has also been demonstrated in SCA7 mouse models. In SCA7 transgenic mice, glial dysfunction leads to the degeneration of neuronal cells by a non-cell-autonomous mechanism via the coordinated action of mutant *ataxin-7* in neurons and glia. Furthermore, the impairment of glutamate transport caused by glial dysfunction contributes to SCA7 neurodegeneration [308,309].

*Drosophila* has an ortholog of human *ataxin-7*. *Drosophila* ataxin-7 plays a conserved role in anchoring a deubiquitinase (Non-stop), which is also a component of the *Drosophila* SAGA complex [310]. Mutations in *non-stop* cause a loss of glial cells, resulting in the misprojection of axons. In *Drosophila*, the loss of ataxin-7 leads to a phenotype that includes aggregation, decreased lifespan, and defective locomotion. This phenotype is due, at least in part, to elevated deubiquitinase activity, since the loss of one copy of *non-stop* suppresses the lethality of ataxin-7 mutants [310]. Within SAGA, the ataxin-7 amino terminus anchors Non-stop to the complex. However, when dissociated, this leads to the subsequent availability of Non-stop for deubiquitination by other substrates, one of which is SCAR, a protein that is essential for actin cytoskeleton organization in *Drosophila* [311,312]. Overexpression of truncated human ataxin-7 with a pathogenic polyQ expansion exhibits similar phenotypes to those due to the loss of ataxin-7 [313,314]. Since Non-stop is a critical mediator of axon guidance and is important for glial cell survival in *Drosophila* [162], polyQ expanded ataxin-7 may play a role in glial dysfunction via Non-stop dysregulation.

## 5. Conclusions and Perspectives

Neurons and glia mutually affect the proper development and normal function of the brain. Many CNS disorders have been extensively studied, but the majority of studies have focused on the events that occur in neurons over long periods. Accumulating evidence has identified glia–neuron interactions in the context of the pathogenesis of these diseases. The recent explosion of data related to *Drosophila* glial cell biology has provided valuable information regarding the role of glia–neuron interactions in the occurrence, as well as the progression, of human CNS disorders. Thus, in this review, we focused on recent advances in the understanding of *Drosophila* glia, exploring key model systems used for the investigation of the basic molecular mechanisms that underlie human CNS disease pathogenesis in six neurodevelopmental and 11 neurodegenerative disorders.

Although in vitro studies provide a powerful approach to exploring this topic, additional extensive in vivo studies are necessary to comprehend the molecular mechanisms underlying glia–neuron communication. For example, the morphology of glial associations with individual neurons has been described in detail; however, the functional significance of these associations or the molecular mechanisms by which the structures have been formed remains unclear. In addition, because of the ease of manipulation, most of our understanding of glia–neuron interactions has been based largely on in vitro studies using primary cultures [315,316]. However, many of the observations of in vitro studies using cultured glia have not been reproduced in intact living organisms [317]. Thus, it would be even more difficult to study in vitro the functional significance of glia–neuron signaling events that have already been disturbed by disease. It is therefore important to utilize suitable in vivo disease models to study human CNS diseases caused by glial malfunction. In addition, as some neurological disorders are characterized by adult onset, the conditional genetic tool of the *Drosophila* disease model is well suited for studying neurodegenerative diseases. For example, the conditional restriction of *TDP-43* (causing ALS10) at the third instar and adult stage leads to abnormal larval movement, aging, and climbing [148]. As another example, the conditional knockdown of APPL at the adult stage was shown to affect night sleep time in AD [142]. In this regard, thus far, *Drosophila* has been an excellent in vivo model system for studying how glial functionality is affected by mutations and how glial dysfunctions contribute to the neuropathogenesis of neurodegenerative disorders as well as neurodevelopmental diseases.

*Drosophila* provides an excellent model for studying disease-associated genes. In most *Drosophila* disease models, a human disease candidate gene can be tested for its potential role using the GAL4/UAS system. Furthermore, the information from such studies can drive research into each disease further forward. EA6, described in Section 4.1.2, is one example where the mechanism behind the disease has been studied in detail using the fly model. Since one EA6-associated mutation, EAAT1^P290R^, is mutated at the residue conserved in *Drosophila*, a *Drosophila* EA6 model can be generated by introducing the UAS construct containing dEAAT1 mutated at the equivalent disease-associated residue (P243R). With this EA6 fly model, it was discovered that the mutation causes the malformation of astrocytes and episodes of paralysis; these phenotypes are rescued by restoring chloride homeostasis to glial cells. This type of in vivo reverse genetic approach to studying variants of human diseases using analogous mutations in fly genes may help to explain the molecular basis of various phenotypes of complicated neurological disorders.

To date, numerous human diseases that are known to be associated with a single gene defect have not been modeled in flies, despite the existence of their orthologs in *Drosophila*. Among them, several cases have not been yet modeled in flies, despite evidence of the association of these neurological disorders with glial dysfunctions in humans. One example of this is the X tremor/ataxia syndrome (FXTAS; OMIM #300623), which is a late onset neurodegenerative disorder [170] caused by a CGG repeat expansion in the premutation range of 55 to 200 repeats in the 5′ non-coding region of *FMR1* [318]. The *Drosophila* ortholog, termed *dFMR1*, has a high degree of amino acid sequence similarity with *FMR1*. The product of *dFMR1* binds RNA and has similar subcellular localization and embryonic expression patterns to mammalian *FMR1* [319]. The neuropathological hallmark of FXTAS patients is intranuclear inclusions throughout the CNS with a highly significant association between the number of CGG repeats and the number of inclusions in both neurons and astrocytes [320,321,322]. Although a FXTAS *Drosophila* model expressing a human *FMR1* premutation allele produces neuron-specific degeneration as well as inclusion bodies similar to those seen in FXTAS patients, the effects of its glial expression have not been reported. It was recently shown that astroglial-targeted expression of the FMRpolyG repeat expansion in mice induces key features of FXTAS pathology, including the formation of intranuclear inclusions, RAN translation, and deficits in motor function [323]. Therefore, an investigation into the response of *Drosophila* glial subtypes to the ectopic expression of human *FMR1* premutation alleles seems intriguing.

Glial dysfunctions are closely associated with the pathogenesis of both neurodevelopmental and neurodegenerative disorders. As reviewed herein, many of these conditions are modeled in *Drosophila*, especially the disorders which are primarily attributed to dysfunctions of a particular factor in glial cells. Thus, to determine possible therapeutic options, the fly model can be utilized for screening pharmaceutical inhibitors of specific targets that counteract the altered functionality that is responsible for diseases. For example, chemical screening of FDA-approved drugs and natural compounds has been performed to identify better treatments for ALXDRD in the ALXDRD fly model. Among 1987 compounds, β-sitosterol, citalopram, duloxetine, and glycopyrrolate function to suppress activated caspase. In particular, glycopyrrolate is an antagonist targeting the mAChR of astrocytes, reducing TUNEL-positive cells and seizures in flies. Inhibition of the mAChR pathway using pharmacologic reagents and genetic approaches reduces GFAP toxicity in the *Drosophila* mutant line. In addition, increased mAChRs and M1 receptors were shown in the astrocytes of a mouse disease model and in a human ALXDRD cell line, respectively [135]. Therefore, *Drosophila* disease models for ALXDRD provide insights into the mechanisms of this disease and especially diseases of glial cell defects. Recent findings from these *Drosophila* models that explore the effects of glial dysfunctions on glia–neuron interactions in the context of human CNS disorders provide insights into the molecular details of glial cell biology and their contributions to disease susceptibility. Thus, although further research is necessary for the compounds screened in the fly models, it will be interesting to establish whether tested compounds that target specific genes associated with glial dysfunction, as outlined in this review, have therapeutic benefits for patients with CNS disorders.

## Figures and Tables

**Figure 1 ijms-21-04859-f001:**
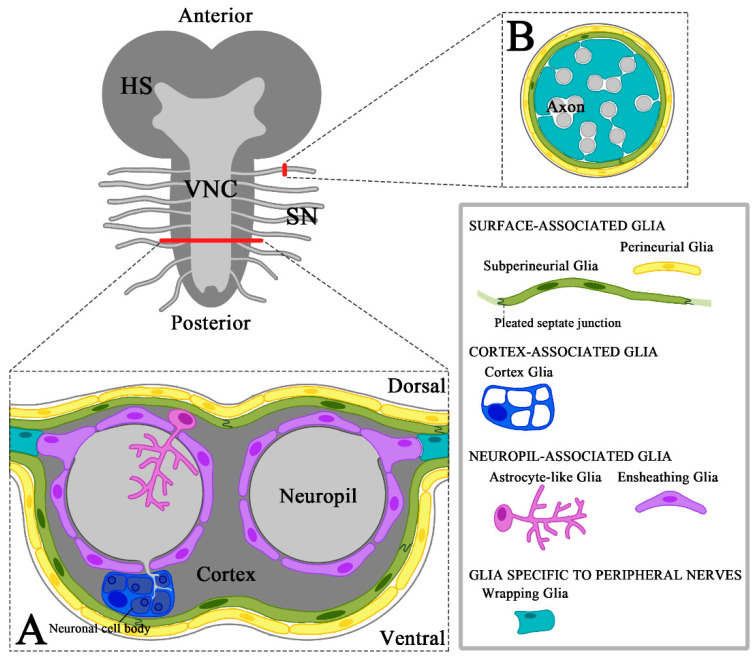
Glial cells of the *Drosophila* larval CNS. The *Drosophila* CNS is composed of two hemispheres (HS) and the ventral nerve cord (VNC). Segmentally organized nerves (SN) connect the VNC to the peripheral nerves. (**A**) Cross-section of the VNC. Glia in the *Drosophila* brain comprise three subtypes: surface-associated glia, cortex-associated glia, and neuropil-associated glia. Surface-associated glia act as a barrier for the CNS. Among them, the subperineurial glial cells form the intercellular pleated septate junction, which blocks paracellular transport and, thus, seals the nervous system from the hemolymph. During larval development, polyploidization takes place in the subperineurial glial cells. Cortex-associated glia wrap all neuronal cell bodies located in the cortex. Ensheathing glia and astrocyte-like glia are included in the neuropil-associated glia. (**B**) Cross-section of the SN at the periphery. Surface-associated glia cover the surface of the SN. Bundles of axons are wrapped by wrapping glia.

**Table 1 ijms-21-04859-t001:** List of CNS disorders summarized in this review and their *Drosophila* models to study the roles of glial defects.

CNS Disorder Types	Disorder (OMIM#; Abbreviation)	Symptom		Human Gene Affected ^†^	Gene Function	*Drosophila* Ortholog ^§^	*Drosophila* Model (Ref.)
**Neuro-developmental disorders**	Alexander disease (#203450; ALXDRD)	Various degrees of macrocephaly, spasticity, ataxia and seizures and leading to psychomotor regression and death		*GFAP*	Intermediate filament protein	-	[133,134,135]
	Episodic ataxia type 6 (#612656; EA6)	Inability to coordinate muscular movements and imbalance, and progressive ataxia		*SLC1A3*	Excitatory amino acid transporter that mediates the uptake of glutamate	*dEAAT1*	[136,137]
	Fragile X syndrome (#300624; FXS)	Cognitive impairment, autistic features, attention deficits, epilepsy, and motor abnormalities		*FMR1*	RNA binding protein in dendrites	*dfmr1*	[138]
	Rett syndrome (#312750; RETT)	Autism, stereotypic hand wiring, respiratory abnormalities, microcephaly, seizures, and mental retardation		*MECP2*	Methyl-CpG-binding protein	-	[139]
	Schizophrenia *	Hallucinations, delusions, thought disorders, and blunted emotions		*DTNBP1*	Member of the biogenesis of lysosome-related organelles complex 1	*Ddysb*	[140]
	SOTOS syndrome 1 (#117550; SOTOS1)	Overgrowth, advanced bone age, macrocephaly, mental retardation, seizures, delayed language, and motor development		*NSD1*	H3K36 methyltransferase	*NSD*	[141]
**Neuro-degenerative disorders**	Alzheimer’s disease 1 (#104300; AD1)	Dementia, synaptic dysfunction, neuronal cell death, impaired learning/memory, and abnormal behaviors		*APP*	Integral membrane protein on the surface of neurons	*dAPPL*	[142,143,144]
	Alzheimer’s disease 2 (#104310; AD2)			*APOE*	Lipoprotein-mediated lipid transport	-	[145]
	Amyotrophic lateral sclerosis 1 (#105400; ALS1)	Death of motor neurons and fatal paralysis		*SOD1*	Cu/Zn superoxide dismutase	*Sod1*	[146]
	Amyotrophic lateral sclerosis 10 (#612069; ALS10)	Frontotemporal lobar degeneration with TDP43 inclusions, death of motor neurons, and fatal paralysis		*TARDBP*	RNA binding protein	*TBPH*	[147,148]
	Ataxia-telangiectasia (#208900; AT)	Cerebellar ataxia, telangiectasia, predisposition to malignancy, and immune disorders		*ATM*	Serine/threonine protein kinase	*dATM*	[149,150]
	Friedreich ataxia (#229300; FA or FRDA)	Progressive gait, limb ataxia, and cardiomyopathy		*FXN*	Biosynthesis of heme and assembly and repair of iron-sulfur cluster	*fh*	[151,152]
	Frontotemporal dementia and/or amyotrophic lateral sclerosis 1 (#105550; FTDALS1)	Shorter survival, bulbar symptom, propensity toward psychosis, and hallucination		*C9* *ORF72*	Component of the guanine nucleotide exchange factor	-	[153]
	Polyglutamine-related disorders	Huntington disease (#143100; HD)	Chorea, cognitive decline, and motor symptoms	*HTT*	Microtubule-mediated transporter or vesicle function	*htt*	[154,155,156]
		Spinocerebellar ataxias type 1 (#164400; SCA1)	Slowly progressive incoordination of gait, poor hand-eye coordination, and dysarthria	*Ataxin-1*	Chromatin-binding factor	*Atx-1*	[157,158]
		Spinocerebellar ataxias type 3 (#109150; SCA3)		*Ataxin-3*	Deubiquitinating enzyme	-	[159,160,161]
		Spinocerebellar ataxias type 7 (#164500; SCA7)		*Ataxin-7*	Component of the STAGA transcription coactivator-HAT complex	-	[162]

* *DTNBP1* is not listed in the schizophrenia phenotype-gene relationships in OMIM; ^†^ HUGO Gene Nomenclature Committee (http://www.genenames.org/); ^§^ denotes the absence of the *Drosophila* ortholog.

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
