# Peer review of "Drosophila Glia: Models for Human Neurodevelopmental and Neurodegenerative Disorders"

_ijms, 2020, doi:10.3390/ijms21144859_

Round 1

Reviewer 1 Report

Manuscript ID: ijms-836655
Type of manuscript: Review
Title: Drosophila glia: models for human neurodevelopmental and neurodegenerative disorders
Authors: Taejoon Kim, Bokyeong Song, Im-Soon Lee

This is a very timely, comprehensive and well-written review. Given the number of human diseases discussed and the frequent use of the Drosophila system in modelling these, this review should be of broad interest. I do not find any major issues to raise, and have only some minor suggestions for improvement.

Rows 71-118: Although not the focus of this review, the generation of glia in the embryo could be covered more comprehensively. Most embryonic glia have been mapped down to NB lineage and single cell identity. It is also known when they are born. I am not sure that their statement that perineurial glia arise at the end of embryogenesis is correct. As far as I know, most glia are born early during neurogenesis i.e., during mid-embryogenesis (Stages 10-14).

Rows 71-118: Relatedly, during larva stages and into pupae, there is a second wave of neurogenesis. This also entails the generation of glia. This wave of glia generation is not at all described.

Rows 133-136: Be sure to state that this refers to mammals.

Rows 161-163: It is not clear that only neurons undergo programmed cell death. Many cells die right after they are born, and hence their identity is not clear. So I would say “…30% of cells die…” Also, for a recent and detailed analysis of the extent of cell death during embryogenesis, please see PMID 28392108.

One major issue pertains to whether or not mutant or misexpression effects on glia represent adult-specific effects, or developmental effects. Most LOF and GOF in the Drosophila field has probably been conducted using non-conditional genotypes, and the effects have often not aligned with whether or not different glia were actually generated and then show phenotype, or perhaps not generated at all, or dies during larval stages. This even relates to a number of transmitter and metabolic enzymes, because a growing literature show that these systems may be important for glia/neuron generation, specification and/or survival. The authors subdivide the human diseases into neurodevelopmental and neurodegenerative, but I am not sure that the Drosophila experiments always have used the proper conditional genetics to actually determine if the effects observed in the fly are developmental and/or adult. Please make sure that it is clearly stated in each section what the Drosophila experiments actually show.

Row 960: It may be worth mentioning that in addition to Abeta-1-42 several other Abeta peptides are generated, which display different levels of toxicity (see e.g. PMID 26208119).

Row 1061: Change “brain” to “CNS”.

Check that Drosophila is in italics throughout, or not, but be consistent.

The review is generally well-organized and well-written, but could nevertheless benefit from some grammatical touch-up. For example:

Row 349: Add “with”.

Row 496: “acts”.

Row 501: add “in”.

Author Response

We, the authors of ijms-836655, appreciate the reviewer’s constructive criticisms on our manuscript, which helped us to revise the manuscript. According to the reviewers’ comments and suggestions, we have made all the corrections in the revised manuscript, and rewritten considerable parts of the text (rewritten paragraphs are shown in red). In addition, we corrected miscellaneous typographical errors as well, and the manuscript has been edited again by a professional English proofreading and editing service. We hope that it now matches the journal's standards. We believe that the contents and clarity of our paper have much improved in the revised version.

We have addressed your concerns point-by-point as follows.

Comment 1: Although not the focus of this review, the generation of glia in the embryo could be covered more comprehensively. Most embryonic glia have been mapped down to NB lineage and single cell identity. It is also known when they are born. I am not sure that their statement that perineurial glia arise at the end of embryogenesis is correct. As far as I know, most glia are born early during neurogenesis i.e., during mid-embryogenesis (Stages 10-14). Relatedly, during larva stages and into pupae, there is a second wave of neurogenesis. This also entails the generation of glia. This wave of glia generation is not at all described. Rows 71-118

Answer: we appreciate and agree with your pertinent comments. As suggested, we more comprehensively described gliogenesis in the embryo and the key genes involved in glial specification. In addition to embryonic glia, we covered morphogenic changes of glial cells during larval and pupal stages, and mentioned the importance of the glial differentiation for the second wave of neurogenesis. Since we stated that all of the Drosophila glia arise during embryogenesis, we omitted the sentence ‘perineurial glia arise at the end of embryogenesis’, which is not only inaccurate but also pointless. Accordingly, we corrected almost the entire paragraph relevantly in the revised manuscript as follows.

Lines 59-64, 75-88

Glial development in Drosophila has been well studied from embryo to adult. Drosophila CNS glial cell progenitors are first formed during embryonic stages. The majority of the embryonic CNS glial cells, the lateral glia, derive from the neurogenic region of the ectoderm [11] while a unique subset of embryonic glia, midline glia, are of mesectoderm origin [12]. The lateral glial cells are produced from neuroglioblasts or glioblasts that generate mixed lineages of neurons and glial cells or glial progeny only, respectively [11].

The lateral glial cells in the Drosophila embryonic CNS have been assigned to three subtypes according to their spatial distribution and morphology: the surface-, the cortex-, and the neuropil-associated glial cells [19]. Although Drosophila gliogenesis begins in the embryo [20,21], important morphogenic changes in glial cells take place during larval and pupal stages. This overlaps with the second wave of neurogenesis responsible for the formation of 90% of the neurons in the adult CNS [22]. For example, the midline glia disappear in the post-embryonic stages [12], while cortex-associated glial cells infiltrate the entire cortex region of the CNS during early larval stages to establish nonoverlapping spatial domains [23]. More drastic events take place during pupal stages. In the case of neuropil-associated glia, the glial cells born during embryonic stages persist until the end of the larval stages, eventually undergoing apoptosis during metamorphosis. Later, adult neuropil-associated glial cells are derived from secondary glia precursor cells that derive from Type II neuroblast lineages during early metamorphosis [24]. Herein, we will limit this review to describe the three main glial subtypes that are found in the CNS of larval and adult Drosophila: surface-associated, cortex-associated, and neuropil-associated glia.

Comment 2: Be sure to state that this refers to mammals. Rows 133-136

Answer: Thank you for your comment. To make it clear, we rephrased the sentence as follows.

Lines 128-131

Mammalian microglia are derived from embryonic mesoderm in the yolk sac [42–44], this contrasts with macroglia (astrocytes and the oligodendrocyte lineage) that arise from neuroepithelial progenitor cells in the embryonic neural tube and forebrain [45].

Comment 3: It is not clear that only neurons undergo programmed cell death. Many cells die right after they are born, and hence their identity is not clear. So, I would say “…30% of cells die…” Also, for a recent and detailed analysis of the extent of cell death during embryogenesis, please see PMID 28392108. Rows 161-163

Answer: We agree with your comment. As suggested, we accordingly corrected the sentence in the revised manuscript. We also appreciate the relevant reference by Cobeta et al. (PMID 28392108), but we did not include the article because the analysis of the PCD during embryogenesis seems a slightly off-topic story for this section.

Lines 152-153

Throughout the embryonic neurogenesis of Drosophila, approximately 30% of all cells undergo programmed cell death and are engulfed by glia [59–62].

Comment 4: One major issue pertains to whether or not mutant or misexpression effects on glia represent adult-specific effects, or developmental effects. Most LOF and GOF in the Drosophila field has probably been conducted using non-conditional genotypes, and the effects have often not aligned with whether or not different glia were actually generated and then show phenotype, or perhaps not generated at all, or dies during larval stages. This even relates to a number of transmitter and metabolic enzymes, because a growing literature show that these systems may be important for glia/neuron generation, specification and/or survival. The authors subdivide the human diseases into neurodevelopmental and neurodegenerative, but I am not sure that the Drosophila experiments always have used the proper conditional genetics to actually determine if the effects observed in the fly are developmental and/or adult. Please make sure that it is clearly stated in each section what the Drosophila experiments actually show.

Answer: we appreciate your pertinent comment. As you mentioned, most of the findings from the Drosophila disease models reviewed in this article are based on non-conditional genotypes, which effects are sometimes difficult to interpret especially if the flies die too early, or disturbing the interactions between neuron and glia due to developmental problems. Thus, the conditional genetics for Drosophila experiments should be chosen as well as analyzed properly. As suggested, we tried to clearly state experimental conditions the Drosophila models in each disease section. In addition, we emphasized again conditional expression (or knockout) cases of two target genes in adult glia among the Drosophila models for neurodegenerative diseases in the ‘conclusion and perspectives’ section as follows.

Lines 819-824

In addition, as some neurological disorders are characterized by adult-onset, the conditional genetic tool of the Drosophila disease model is well suited for studying neurodegenerative disease. For example, conditional restriction of TDP-43 (causing ALS10) at the third instar and adult stage leads to abnormal larval movement, aging, and climbing [214]. As another example, conditional knockdown of APPL at the adult stage was shown to affect night sleep time in AD [192].

Comment 5 : It may be worth mentioning that in addition to Abeta-1-42 several other Abeta peptides are generated, which display different levels of toxicity (see e.g. PMID 26208119). Row 960

Answer: Thank you for your comment. As suggested, we added description on other toxic Aβ peptides by referring to the paper by Jonson et al. (PMID:26208119).

Lines 464-470

Furthermore, ectopic expression of human Aβ42 peptides in the fly brain causes age-dependent learning defects and short-term memory impairment due to their elevated aggregation propensity. In addition to Aβ42, several other Aβ peptides that display different levels of toxicity as well as aggregation have been reported [191]. Jonson et al. (2015) demonstrated that the neuronal expression of Aβ42 peptides that are mutated at the N- or C-terminal differentially contribute to Aβ toxicity. N-terminal truncated peptides present a less severe phenotype but are quite toxic, while mutating the C-terminal residue 42 in Aβ42 greatly reduces Aβ accumulation and toxicity.

Comment 6: Change “brain” to “CNS”. Row 1061

Answer: Thank you for your comment. The word ‘brain’ is changed to ‘CNS’.

Lines 892-893

Glial cells of the Drosophila larval CNS. The Drosophila CNS is composed of two hemispheres (HS) and the ventral nerve cord (VNC).

Comment 7: Check that Drosophila is in italics throughout, or not, but be consistent.

Answer: Thank you for your comment. We did double-check if the word ‘Drosophila’ is in italics.

Comment 8: The review is generally well-organized and well-written, but could nevertheless benefit from some grammatical touch-up. For example: Row 349: Add “with”. Row 496: “acts”. Row 501: add “in”.

Answer: Thank you for your comment. The manuscript has been edited again by a professional English proofreading and editing service, and we corrected the sentence as suggested.

Line 384-385; 428-429; 432-434

In Drosophila, MeCP2 is phosphorylated at serine 423 and interacts with sin3a, N-coR, and REST.

The Drosophila NSD1 ortholog, NSD, interacts with various chromatin remodeling factors such as HP1 and insulator binding protein dCTCF/Beaf32, acting downstream of the DRE/DREF pathway [177–179]

Interestingly, NSD overexpression in glial cells, not neurons, induces brain cell death, behavioral defects, and a decrease in brain size that is similar to the microcephaly phenotype of patients with NSD1 duplication.

-----------------------------------------------------------------------

We hope that we have provided appropriate and satisfying responses to the reviewer’s comments and questions.

We look forward to receiving a positive response from you.

Im-Soon Lee

Professor

Department of Biological Sciences

Konkuk University, Seoul 05029, South Korea

Phone: 82-2-450-4213, Fax: 82-2-3436-5432, e-mail: islee@konkuk.ac.kr

Reviewer 2 Report

This is an extensive summary of historical and more recent attempts to use Drosophila to model human neurodevelopmental and neurodegenerative diseases, specifically the involvement of glial cells in those processes. The manuscript first introduces Drosophila glial cells and their functions before going on to discuss details of modelling of neurodevelopmental and neurodegenerative diseases one-by-one. An overview of the relevant human genetics, manifestations of the disease is introduced before Drosophila-centric experiments are described. I enjoyed reading the review and it is timely given that the last update in this area was a relatively short summary provided by Mary Logan (“Glial Contributions to Neuronal Health and Disease: New Insights From Drosophila” - Curr Op Neurobiol, 2017). Prior to that the most recent/specific review appears to be "Drosophila as a model to study the role of glia in neurodegeneration" (Lee YM and Sun YH. J Neurogenet. 2015;29(2-3):69-79. doi: 10.3109/01677063.2015.1076816.)

This excellent summary updates the field and is useful both to investigators with an interest in Drosophila glial cells and/or disease modellers but also more broadly to those with an interest in these areas but who predominantly use other models. I have some suggestions on how the review could be made of more broad interest (see below).

The review is largely well written but some sections could be made more clear and the manuscript needs careful checking and copy-editing. To help the authors I have highlighted some mistakes I have found (I hope this is helpful and not patronising!) I have highlighted the most culpable sections below.

The introduction is short and to the point and sections 2 and 3, which introduce glial subtypes and their function are also suitably succinct. This enables the bulk of the article to concentrate on the Drosophila models used to understand neurodevelopment and degeneration. Section 2 is slightly longer and I like how relevant processes are related back to vertebrate glial populations.

Overall this is a useful primer and update for the field and I support its publication once some relatively minor issues have been addressed (though I have grouped my comments into major and minor points based on how much effort is required to improve the manuscript).

Major comments

-------------------

1. A description of the key genes involved in specification (repo, gcm, gcm2, tramtrack, pointed) is missing and should probably be considered (briefly - there are numerous reviews on this topic). However, explaining what Repo is necessary as a minimum as "Repo drivers" are used in a subsequent section without sufficient explanation for the non-expert.

Furthermore both recent and more historical glial profiling work (e.g. work by groups of Giangrande/Freeman/Egger) may be useful to mention as this provides  molecular details that may be useful to scientists interested in this area (e.g. lists of genes to facilitate further modelling). The Freeman paper I am thinking of is cited [49] but not in this context.

2. I don’t understand why the neurodevelopment section starts with RETT/MECP2, a model in which flies lack an orthologue. This section would be better to start with those with the most clear rationale for use in modelling of human disease. How has section 4.1 been ordered? Could it be ordered according to gain of function approaches and (the more relevant and useful) models in which a clear orthologue is present?

3. The descriptions of disease modelling are excellent, but what would be useful would be to be more explicit and explain how findings in the fly have driven research into each disease further forward (if this is the case), as opposed to simply reciprocating molecular players and mechanisms involved. This has been done in some cases (e.g. EA6), but there is no comment on whether these ideas/findings have been taken on/confirmed by those using vertebrates. This could alternatively be provided in the perspectives section at the end of the disease modelling section. Adding some small detail in this way may help convince more of the utility of the model and make it of greater interest to a broader swathe of researchers.

4. Section 4.1.4: why is the association of DISC1 with schizophrenia discussed when this section goes on to concentrate on dysbindin? The discussion of DISC1 should be curtailed and replaced with (or integrated with, if there are similar/related mechanisms) a discussion of what is known about dysb.

5. Section 4.2.5 is quite complex and could be re-written to make it clearer. I appreciate this is a complex and poorly understood disease, but some improvements would help the reader, especially those without a background in this area.

6. Can the authors comment on which diseases have not been modelled in flies and which may be able to be modelled based on the existence of homologues?

7. The manuscript requires careful copy editing - some of the genes and abbreviations are incorrect and there is the occasional grammatical/typographical mistake.

Minor comments (typos etc)

-------------------------------

Figure 1 - category for wrapping glia (Glia appeared only in the peripheral nerves) should be re-written as something like Glia specific to peripheral nerves or similar

Section 3.1, line 126 “suggesting an increase in function alongside the increasing evolutionary complexity of the nervous system” - this could be written more clearly - a diversification in the roles carried out?

Line 133 - make explicit these are mammalian microglia.

Line 138 - please provide examples of molecules in common between fly and vertebrate glia.

Line 204 - swiss cheese should be in italics as per convention and brief detail of actual molecular function of swiss cheese here would be helpful (and if it relates to conserved genes/processes in vertebrates).

Line 238-239 - reference detailing missing for recent MCT’s identification.

line 368 - Large expansions rather than largely? I.e. do the authors mean repeats are responsible or do they wish to imply other mechanisms can contribute?

line 437 - missing “the” before mouse brain.

Line 485-487 this sentence reads slightly oddly (missing morphology/structure after face? phenotype at the end?)

Line 601 - relevance of repo drivers not explained (first mention of repo), which would make it hard for a non-Drosophilist to follow this section. Perhaps a brief discussion of drivers and key transcription factors could be incorporated in the subtypes of glial cell section.

line 686 - misspelling of ataxin7.

line 882 DRP rather than DPR (also lines 896 & 900). This section (4.2.5) may require some editing as writing could be clearer in places (it gets quite complex with all the cell autonomies, for instance).

line 978 - Drosophila not italicised, 979 missing “a” before defective.

line 1020 - change “are” to “can be”.

Author Response

We, the authors of ijms-836655, appreciate the reviewer’s constructive criticisms on our manuscript, which helped us to revise the manuscript. According to the reviewers’ comments and suggestions, we have made all the corrections in the revised manuscript, and rewritten considerable parts of the text (rewritten paragraphs are shown in red). In addition, we corrected miscellaneous typographical errors as well, and the manuscript has been edited again by a professional English proofreading and editing service. We hope that it now matches the journal's standards. We believe that the contents and clarity of our paper have much improved in the revised version.

We have addressed your concerns point-by-point as follows.

Comment 1: A description of the key genes involved in specification (repo, gcm, gcm2, tramtrack, pointed) is missing and should probably be considered (briefly - there are numerous reviews on this topic). However, explaining what Repo is necessary as a minimum as "Repo drivers" are used in a subsequent section without sufficient explanation for the non-expert. Furthermore, both recent and more historical glial profiling work (e.g. work by groups of Giangrande/Freeman/Egger) may be useful to mention as this provides molecular details that may be useful to scientists interested in this area (e.g. lists of genes to facilitate further modelling). The Freeman paper I am thinking of is cited [49] but not in this context.

Answer: Thank you for your comment. In embryonic neurogenesis, Gcm and Gcm2 are the major transcriptional factors for neuroblast differentiation. Their target genes (repo, loco, pnt, and ttk) play crucial roles for glia and neuronal cell fates. Thus, we included description on molecular details associated with the key genes functioning during glial specification as follows.

Lines 64-75

In the presence of the glial cell missing (Gcm) transcription factor and its related factor, Gcm2, which act as fate determinants, the lateral glial cells are specified according to positional information from the neuroectoderm [13]. In the lateral glial cells, Gcm activates a set of downstream transcription factors that control the differentiation and maintenance of the glial cell fate. Among the target genes of Gcm, the reversed polarity (Repo) homeodomain transcription factor promotes the proteasome-mediated degradation of Gcm and positively regulates its own promoter, resulting in its sustained expression in glia. Together with Repo, other Gcm-induced transcriptional factors also play crucial roles in glial specification, such as locomotion defects (Loco), pointed (Pnt), and tramtrack (Ttk). Pnt and Loco function as activators of glial fate [14–17] while Ttk is a repressor of neuronal fate [18]. Thus, neuroblasts from the neuroectoderm appear to have an inherent primary fate to develop as neurons, and a neuronal fate is chosen in the absence of Gcm.

Comment 2: I don’t understand why the neurodevelopment section starts with RETT/MECP2, a model in which flies lack an orthologue. This section would be better to start with those with the most clear rationale for use in modelling of human disease. How has section 4.1 been ordered? Could it be ordered according to gain of function approaches and (the more relevant and useful) models in which a clear orthologue is present?

Answer: Thank you for your comment. However, we realized that description on some disease cases (ex. polyQ diseases or SCA) become more complicated when classified into those with and without the ortholog. Thus, we simply reorganized the diseases in the alphabetical order in the text and the table as follows.

Neurodevelopmental disease

Alexander disease; Episodic ataxia type 6; Fragile X syndrome; Schizophrenia; SOTOS syndrome 1;

Rett syndrome

Neurodegenerative disease

Alzheimer’s disease 1,2; Amyotrophic lateral sclerosis 1; Amyotrophic lateral sclerosis 10;

Ataxia-telangiectasia Fragile X tremor/ataxia syndrome; Friedreich ataxia;

Frontotemporal dementia and/or amyotrophic lateral sclerosis 1; Huntington disease

Spinocerebellar ataxias type 1,3,7

Comment 3: The descriptions of disease modelling are excellent, but what would be useful would be to be more explicit and explain how findings in the fly have driven research into each disease further forward (if this is the case), as opposed to simply reciprocating molecular players and mechanisms involved. This has been done in some cases (e.g. EA6), but there is no comment on whether these ideas/findings have been taken on/confirmed by those using vertebrates. This could alternatively be provided in the perspectives section at the end of the disease modelling section. Adding some small detail in this way may help convince more of the utility of the model and make it of greater interest to a broader swathe of researchers.

Answer: Thank you for your comment. As you mentioned, Drosophila provides an excellent model to drive research further forward into vertebrate studies. To emphasize this matter, we added some details on the fly EA6 model case in the conclusion and perspectives section as follows.

Lines 828-839

Drosophila provides an excellent model for studying disease-associated genes. In most Drosophila disease models, a human disease candidate gene can be tested for its potential role using the GAL4/UAS-system. Furthermore, the information from such studies can drive research into each disease further forward. EA6, described in the section 4.1.2, is one example where the mechanism behind the disease has been studied in detail using the fly model. Since one EA6-associated mutation, EAAT1P290R, is mutated at the residue conserved in Drosophila, a Drosophila EA6 model can be generated by introducing the UAS construct containing dEAAT1 mutated at the equivalent disease-associated residue (P243R). With this EA6 fly model, it was discovered that the mutation causes malformation of astrocytes and episodes of paralysis; these phenotypes are rescued by restoring chloride homeostasis to glial cells. This type of in vivo reverse genetic approach to studying variants of human diseases using analogous mutations in fly genes may help to comprehend the molecular basis of various phenotypes of complicated neurological disorders.

Comment 4: Section 4.1.4: why is the association of DISC1 with schizophrenia discussed when this section goes on to concentrate on dysbindin? The discussion of DISC1 should be curtailed and replaced with (or integrated with, if there are similar/related mechanisms) a discussion of what is known about dysb.

Answer: We totally agree to your opinion. We previously described SCZD9 in the schizophrenia section because the DISC1 gene is well-established as a single candidate gene for this disease, and its SNP has been reported to be associated with defects in astrogenesis with abnormal behaviors in the mouse model. However, since mutant DISC1 expression as well as its effects in Drosophila glia has not been reported to date, we now more focus on the description of SCZD3 associated with variations in the DTNBP1 gene in the revised manuscript, as suggested.

Lines 400-407

Schizophrenia appears to be highly heritable, but the genetics are complex [166]. There may not be a single cause; numerous SCZD susceptibility genes have been identified thus far. Straub et al. (2002) were the first to report that SNPs within the gene for dystrobrevin-binding-protein 1 (DTNBP1), also known as dysbindin (dysb), were strongly associated with SCZD [167]. This gene is currently considered one of the candidate genes for SCZD3 (OMIM %600511). DTNBP1 is a component of the biogenesis of lysosome-related organelles complex 1 and plays a role in glutamate neurotransmission. It is widely expressed in the brain [168,169], and a role of glial DTNBP1 expression in SCZD has been suggested in a Drosophila model.

Comment 5: Section 4.2.5 is quite complex and could be re-written to make it clearer. I appreciate this is a complex and poorly understood disease, but some improvements would help the reader, especially those without a background in this area.

Answer: Thank you for your comment, and we re-wrote the section to make it clearer, as mentioned in the comment 19.

Comment 6: Can the authors comment on which diseases have not been modelled in flies and which may be able to be modelled based on the existence of homologues?

Answer: Thank you for your comment. Drosophila disease model are well established for a number of neurological diseases based on the existence of human homologs. However, although the number of reports supporting the association of neurological diseases with glia dysfunction in mammalian models is increasing, only a few diseases have been modelled in Drosophila thus far as listed in this review. Thus, we described a neurological disease as an example that has been reported to be associated with glial dysfunction so that it might be possibly modelled in Drosophila as follows.

Lines 840-857

To date, numerous human diseases that are known to be associated with a single gene defect have not been modelled in flies despite the existence of their orthologs in Drosophila. Among them, several cases have not been yet modelled in flies despite evidence of the association of these neurological disorders with glial dysfunctions in humans. One example of this is the X tremor/ataxia syndrome (FXTAS; OMIM #300623), which is a late-onset neurodegenerative disorder [144] caused by a CGG repeat expansion in the premutation range of 55 to 200 repeats in the 5′ non-coding region of FMR1 [317]. The Drosophila ortholog, termed dFMR1, has a high degree of amino acid sequence similarity with FMR1. The product of dFMR1 binds RNA and has similar subcellular localization and embryonic expression patterns to mammalian FMR1 [318]. The neuropathological hallmark of FXTAS patients is intranuclear inclusions throughout the CNS with a highly significant association between the number of CGG repeats and the number of inclusions in both neurons and astrocytes [319–321]. Although a FXTAS Drosophila model expressing a human FMR1 premutation allele produces neuron-specific degeneration as well as inclusion bodies similar to those seen in FXTAS patients, the effects of its glial expression have not been reported. It was recently shown that astroglial-targeted expression of the FMRpolyG repeat expansion in mice induces key features of FXTAS pathology, including formation of intranuclear inclusions, RAN translation, and deficits in motor function [322]. Therefore, investigation into the response of Drosophila glial subtypes to ectopic expression of human FMR1 premutation alleles seems intriguing.

Comment 7: The manuscript requires careful copy editing - some of the genes and abbreviations are incorrect and there is the occasional grammatical/typographical mistake.

Answer: Thank you for your comment. The manuscript has been edited again by a professional English proofreading and editing service.

Comment 8: Figure 1 - category for wrapping glia (Glia appeared only in the peripheral nerves) should be re-written as something like Glia specific to peripheral nerves or similar

Answer: Thank you for your comment. As suggested, category for wrapping glia has been rewritten as ‘Glia specific to peripheral nerves’ for clarity.

Comment 9: Section 3.1, line 126 “suggesting an increase in function alongside the increasing evolutionary complexity of the nervous system” - this could be written more clearly - a diversification in the roles carried out?

Answer: Thank you for your comment. As suggested, we corrected the sentence more clearly as follows.

Lines 119-123

The glial cells in the Drosophila nervous system are considerably simpler but distinct from their mammalian counterparts. For example, Drosophila glia comprise only approximately 10–15% of the 90,000 cells estimated to arise in the adult CNS, whereas mammals contain more glia: 50% in mice and 90% in humans. This suggests an expansion of glial roles, such as learning, as the complexity of the nervous system increases [3–5,40].

Comment 10: Line 133 - make explicit these are mammalian microglia.

Answer: As suggested, the sentences have been corrected and rephrased as follows.

Lines 128-131

Mammalian microglia are derived from embryonic mesoderm in the yolk sac [42–44], this contrasts with macroglia (astrocytes and the oligodendrocyte lineage) that arise from neuroepithelial progenitor cells in the embryonic neural tube and forebrain [45].

Comment 11: Line 138 - please provide examples of molecules in common between fly and vertebrate glia.

Answer: Thank you for your comment. We added K+ channels as an example, which is conserved from fly to vertebrate, as follows.

Lines 133-134

Molecules mediating electrical excitability and related channels in Drosophila are generally similar to those in humans, including voltage-activated K+ channels [47].

Comment 12: Line 204 - swiss cheese should be in italics as per convention and brief detail of actual molecular function of swiss cheese here would be helpful (and if it relates to conserved genes/processes in vertebrates).

Answer: Thank you for your comment. Mutants of the swiss cheese (sws) gene in Drosophila induce neuronal apoptosis and glial hyperwrapping. The sws gene is the Drosophila ortholog of NTE. Thus, we added the details of the sws gene and rewrote the gene in italic accordingly, as follows.

Lines 188-193

The mutations in sws develop axonal and glial pathology and neuronal apoptosis. Additionally, the SWS protein was recently shown to play a role in the maintenance of neuromuscular function development and microtubule networks [88]. The sws gene is the ortholog of the neuropathy target esterase (NTE) gene, which is one of the genetic factors responsible for the development of hereditary spastic paraplegia [88].

Comment 13: Line 238-239 - reference detailing missing for recent MCT’s identification.

Answer: Thank you for your comment. As suggested, we added reference (PMID : 29352169).

Lines 221-222

Recently, the presence of a functionally active MCT was identified in the Drosophila CNS, implying its role in intercellular lactate shuttling in the Drosophila brain [103].

Comment 14: line 368 - Large expansions rather than largely? I.e. do the authors mean repeats are responsible or do they wish to imply other mechanisms can contribute?

Answer: Thank you for your comment. As suggested, we provided more precise repeat number affecting FXS as follows.

Lines 339-342

More than 200 CGG repeats in the 5’ untranslated region (UTR) of the fragile X mental retardation 1 gene (FMR1, also known as the gene for FMRP translational regulator 1) are responsible for FXS. The CGG repeats in the 5’ UTR induce DNA methylation of FMR1, which leads to transcriptional silencing [146].

Comment 15: line 437 - missing “the” before mouse brain.

Answer: For the reason mentioned in the Comment #4, the whole paragraph has been omitted. Thank you for pointing out.

Comment 16: Line 485-487 this sentence reads slightly oddly (missing morphology/structure after face? phenotype at the end?

Answer: Thank you for your comment. As suggested, we corrected the sentence as follows.

Lines 418-419

Sotos syndrome 1 (SOTOS1; OMIM #117550) is a developmental disorder characterized by facial abnormalities, advanced bone age, and macrocephaly.

Comment 17: Line 601 - relevance of repo drivers not explained (first mention of repo), which would make it hard for a non-Drosophilist to follow this section. Perhaps a brief discussion of drivers and key transcription factors could be incorporated in the subtypes of glial cell section.

Answer: Thank you for your comment. We described the relevance of the repo driver as follows. We also mentioned the role of the Repo protein as a transcriptional activator in the section 2 as described in the Comment #1 of Reviewer #2.

Lines 726-729

Furthermore, when expressed by the pan-glial cell specific Repo drivers, the developmental effect of the expanded ataxin-1 on the earliest stages of glial development is more prominent than that at later stages. This suggests a possible role for glial polyQ expanded ataxin-1 pathology in early onset of the disease.

Comment 18: line 686 - misspelling of ataxin7.

Answer: Thank you for your comment. We corrected the misspelling of the gene in this sentence and throughout the section.

Lines 797-799

Since Non-stop is a critical mediator of axon guidance and is important for glial cell survival in Drosophila [313], polyQ expanded ataxin-7 may play a role in glial dysfunction via Non-stop dysregulation.

Comment 19: line 882 DRP rather than DPR (also lines 896 & 900). This section (4.2.5) may require some editing as writing could be clearer in places (it gets quite complex with all the cell autonomies, for instance).

Answer: Thank you for your comment. However, we changed DRP to ‘DPR proteins’ for the dipeptide repeat proteins instead to avoid confusion between the two abbreviations. As for the cell autonomies, since the interaction between neuron and glia can be explained either by cell-autonomous or by non-cell-autonomous manners, it is difficult to explain their interaction without using them. Thus, instead of substitution of the words, we removed the words, ‘cell-autonomous’ or ‘non-cell-autonomous’, if unnecessary to avoid complication. In addition, we restructured the section to make it clear.

Lines 621-650

In studies using FTDALS1 patient-derived cells, C9orf72 HRE was shown to reduce C9ORF72 expression, causing neurodegeneration via the accumulation of glutamate receptors, along with impaired clearance of neurotoxic dipeptide repeat (DPR) proteins [250]. In addition, using human iPSC-derived astrocytes, astrocytic C9orf72 mutation was confirmed to be responsible for both cell-autonomous astrocyte pathology and non-cell-autonomous motor neuron pathophysiology [251].

C9ORF72 mRNA and its protein are highly expressed in normal mice and human brains; loss-of-function homozygous mutations in C9orf72 lead to premature death in mice [252,253]. However, neural‐specific ablation of C9orf72 in the conditional knockout mice is insufficient to cause motor neuron disease [254]. Despite a recent report showing reduced C9ORF72 function increases C9orf72 HRE toxicity [255], the gain-of-function of C9orf72 HRE has been known to play more crucial roles in neurodegenerative changes. For this, two molecular mechanisms have been suggested: toxicity from HRE-containing RNA and the accumulation of toxic DPR proteins via RAN translation [256].

Compared to the crucial roles of neurons in motor and behavioral phenotypes in FTDALS1 identified by mouse models expressing C9orf72 DPR proteins [257,258], few studies have demonstrated a direct glial contribution to FTDALS1 neuropathology [250]. Recently, DPR proteins in transgenic mice were shown to non‐cell‐autonomously trigger key features of FTDALS1, including cytoplasmic mislocalization and aggregation of TDP‐43, suggesting a possible role of glial cells as neighboring cells of degenerating neurons [259].

C9orf72 is highly conserved in evolution [260], but no Drosophila ortholog has been identified. However, Drosophila has been widely used to model C9orf72 HRE RNA and DPR protein toxicity through the generation of transgenic flies. In an FTDALS1 Drosophila model, the C9orf72 DPR protein causes neurodegenerative phenotypes and excitotoxicity in glutamatergic neurons [261]. Using novel Drosophila FTDALS1 models that produce spliced intronic nuclear HRE RNA or poly(A)+ DPR mRNA exported to the cytoplasm for protein production, it was revealed that toxicity is correlated with HRE-derived DPR protein production but not HRE-RNA accumulation [262]. In addition, a vicious feedback loop was recently found in C9orf72 DPR transgenic flies, in which DPR proteins, but not HRE RNA accumulation, regulate TDP-43 dysfunction and TDP-43 inversely increases levels of DPR proteins [263].

Comment 20: line 978 - Drosophila not italicised, 979 missing “a” before defective.

Answer: Thank you for your comment. As suggested, we corrected the sentence, as follows.

Lines 485-487

In a human APOE transgenic Drosophila model, the E4 variant exhibits a defective response to oxidative stress, resulting in increased neurotoxicity, whereas the E3 variant has a neuroprotective effect [197].

Comment 21: line 1020 - change “are” to “can be”

Answer: Thank you for your comment. As suggested, we corrected the sentence, as follows.

Lines 861-863

Thus, to determine possible therapeutic options, the model fly can be utilized for screening pharmaceutical inhibitors of specific targets that counteract the altered functionality responsible for diseases.

-----------------------------------------------------------------------

We hope that we have provided appropriate and satisfying responses to the reviewer’s comments and questions.

We look forward to receiving a positive response from you.

Im-Soon Lee

Professor

Department of Biological Sciences

Konkuk University, Seoul 05029, South Korea

Phone: 82-2-450-4213, Fax: 82-2-3436-5432, e-mail: islee@konkuk.ac.kr

Round 2

Reviewer 2 Report

The authors have extensively addressed all of my comments/concerns and I recommend that this comprehensive review is now accepted.

I recommend that the authors double check that genes are in italics and cited as per conventional genetic nomenclature (see Flybase). For example, in comment 1, authors refer to gene targets of Gcm but these are not in italics.